# Chromatin modifier Hmga2 promotes adult hematopoietic stem cell function and blood regeneration in stress conditions

Sho Kubota [1,2], Yuqi Sun [1,3], Mariko Morii [1], Jie Bai [1], Takako Ideue[1], Mayumi Hirayama[1], Supannika Sorin [1], Eerdunduleng[1], Takako Yokomizo-Nakano [1], Motomi Osato[1,4], Ai Hamashima [1], Mihoko Iimori[1], Kimi Araki[5,6], Terumasa Umemoto [7] & Goro Sashida [1✉]

## Abstract

The molecular mechanisms governing the response of hematopoietic stem cells (HSCs) to stress insults remain poorly defined. Here, we investigated effects of conditional knock-out or overexpression of *Hmga2* (High mobility group AT-hook 2), a transcriptional activator of stem cell genes in fetal HSCs. While *Hmga2* overexpression did not affect adult hematopoiesis under homeostasis, it accelerated HSC expansion in response to injection with 5-fluorouracil (5-FU) or in vitro treatment with TNF-α. In contrast, HSC and megakaryocyte progenitor cell numbers were decreased in *Hmga2* KO animals. Transcription of inflammatory genes was repressed in *Hmga2*-overexpressing mice injected with 5-FU, and Hmga2 bound to distinct regions and chromatin accessibility was decreased in HSCs upon stress. Mechanistically, we found that casein kinase 2 (CK2) phosphorylates the Hmga2 acidic domain, promoting its access and binding to chromatin, transcription of anti-inflammatory target genes, and the expansion of HSCs under stress conditions. Notably, the identified stress-regulated Hmga2 gene signature is activated in hematopoietic stem progenitor cells of human myelodysplastic syndrome patients. In sum, these results reveal a TNF-α/CK2/phospho-Hmga2 axis controlling adult stress hematopoiesis.

**Keywords** TNF-α; Casein Kinase 2; 5-Fluorouracil; Self-renewal; Megakaryocyte
**Subject Categories** Chromatin, Transcription & Genomics; Haematology; Stem Cells & Regenerative Medicine

## Introduction

Tissue stem cells have a self-renewal capacity and multipotency to maintain adult tissue homeostasis and repair stress-induced tissue damage (Wagers and Weissman, 2004; Goodell and Rando, 2015). Stress is the disruption of homeostasis through changes in signaling, epigenomic, and transcriptional mechanisms mediated by extrinsic stimuli and intrinsic factors, resulting in transient or irreversible tissue damage, such as aging-associated diseases including cancer (Oh et al, 2014; López-Otín et al, 2023). Hematopoietic stem cells (HSCs) respond to various stresses, such as infection, inflammation, and myeloablation, and expand hematopoietic stem and progenitor cells (HSPCs) to produce mature blood cells (Essers et al, 2009; Baldridge et al, 2010; Haas et al, 2015); however, the mechanisms by which HSCs maintain hematopoiesis based on their self-renewal and differentiation fates in differential responses to homeostatic and stress conditions have yet to be elucidated.

High mobility group AT-hook 2 (Hmga2), a chromatin modifier protein, is highly expressed in fetal HSCs and is necessary for an enhanced self-renewal capacity (Copley et al, 2013). Hmga2 activates the transcription of target genes by competing with the linker histone H1 and/or binding to the regulatory regions of genes via its AT-hook domains (Fusco and Fedele, 2007). The overexpression of *Hmga2* via a retrovirus vector or a transgene was shown to enhance the self-renewal capacity of adult HSCs and the megakaryocyte and erythroid differentiation in humans and mice under in vivo conditions (Copley et al, 2013; Rowe et al, 2016; Kumar et al, 2019), but was insufficient to induce leukemic transformation (Ikeda et al, 2011; Bai et al, 2021). We generated *Rosa26* locus *Hmga2* conditional knock-in (KI) mice and recently reported that the overexpression of *Hmga2* enhanced the self-renewal capacity of adult HSCs and maintained their fitness in the bone marrow (BM) of both primary mice and transplanted mice (Sun et al, 2022), while lethal irradiation followed by BM

[1]Laboratory of Transcriptional Regulation in Leukemogenesis, International Research Center for Medical Sciences, Kumamoto University, Kumamoto, Japan. [2]Department of Medicinal Pharmacology, Graduate School of Medicine, Dentistry and Pharmaceutical Sciences, Okayama University, Okayama, Japan. [3]Department of Hematology, The Second Affiliated Hospital of Fujian Medical University, Quanzhou, Fujian, China. [4]Department of General Internal Medicine, Kumamoto Kenhoku Hospital, Kumamoto, Japan. [5]Institute of Resource Development and Analysis, Kumamoto University, Kumamoto, Japan. [6]Center for Metabolic Regulation of Healthy Aging, Kumamoto University, Kumamoto, Japan. [7]Laboratory of Hematopoietic Stem Cell Engineering, International Research Center for Medical Sciences, Kumamoto University, Kumamoto, Japan. ✉E-mail: sashidag@kumamoto-u.ac.jp

transplantation induced the excessive replication of HSCs and systemic inflammation and subsequently depleted wild-type (WT) HSCs in the long term (Rodrigues-Moreira et al, 2017; Caiado et al, 2021).

In the present study, we generated new *Hmga2* conditional knock-out (cKO) mice and examined normal and stress hematopoiesis using both *Hmga2* cKO and KI mice. Under homeostatic conditions, neither *Hmga2* cKO nor KI mice showed a significant change in hematopoietic phenotypes; however, in response to an injection of 5-fluorouracil (5-FU), the proliferation of HSCs was enhanced in *Hmga2* KI mice and hematopoietic recovery was rapid, whereas HSC and megakaryocyte progenitor (MkP) cell numbers significantly decreased in *Hmga2* cKO mice and hematopoietic recovery was delayed. Since the Hmga2 gene appeared to repress the transcription of interferon-stimulated genes and inflammatory response genes in HSCs, we assessed Hmga2-binding regions and chromatin accessibility in cells after the in vivo injection of 5-FU or an in vitro treatment with TNF-α, a critical cytokine in inflammatory responses (Pronk et al, 2011; Yamashita and Passegué, 2019). We found that Hmga2 bound to distinct regions and reduced chromatin accessibility in inflammation regulator genes, such as the Rfx5 transcription factor (TF), after these stress insults. Since the acidic domain in the Hmga2 protein was directly phosphorylated by Casein kinase 2 (CK2) (Sgarra et al, 2009), which has been implicated in the modulation of the TNF-α-NF-kB pathway (Borgo et al, 2021), the inhibition of phosphorylation in the acidic domain by a substitution with alanine impaired the stress-induced chromatin binding of Hmga2, chromatin accessibility, the transcription of target genes, and the expansion of HSCs under stress conditions. While the *Hmga2* gene did not appear to be necessary for adult hematopoiesis under homeostatic conditions, we herein elucidated the molecular mechanisms by which HSCs drove hematopoietic regeneration under stress conditions by activating the expression and function of Hmga2.

## Results

### The overexpression of Hmga2 enhanced the expansion of HSCs and production of erythroid and MkP cells under stress conditions

We previously reported a *Rosa26-flox-stop-flox-HA-Hmga2-eGFP* conditional KI (*Hmga2* KI) mouse showing the inducible overexpression of *Hmga2* mRNA in adult HSCs to a similar degree as that in fetal HSCs (Sun et al, 2022). A serial transplantation assay revealed that the overexpression of *Hmga2* enhanced the competitive repopulating capacity of HSCs, but did not confer that property to multipotent progenitor (MPP) cells, while WT cells completely lost their repopulation capacity after transplantation (Fig. EV1). Based on the enhanced self-renewal of *Hmga2* KI HSCs under transplantation stress, we examined the response of *Hmga2* KI HSCs to myeloablation induced by an in vivo treatment with 5-FU. We intravenously injected 250 mg/kg 5-FU into control WT mice and *Hmga2* KI mice (Fig. 1A). Under inflammatory conditions, we defined HSCs as Lineage⁻EPCR⁺CD48⁻CD150⁺, which mostly expressed c-Kit and Sca-1 (Fig. EV2), based on our previous finding on this phenotypic definition of functional HSCs (Umemoto et al, 2022). This single injection of 5-FU was shown to induce severe damage to normal

hematopoiesis in mice that persisted for more than three weeks (Umemoto et al, 2022), while *Hmga2* KI mice showed the quicker recovery of the complete blood cell count with higher hemoglobin levels and platelet counts on day 12 after the injection of 5-FU than those in WT mice (Fig. 1B). *Hmga2* KI mice simultaneously showed higher BM cell counts and more HSCs on days 7 and 12 after the injection of 5-FU than those in WT mice (Fig. 1C,D). Consistent with the rapid recoveries of hemoglobin levels and platelet counts in *Hmga2* KI mice, flow cytometric analyses revealed that *Hmga2* KI mice contained more megakaryocyte-biased CD41⁺ HSCs, erythroid progenitor cells, and MkP cells in BM on day 12 after the 5-FU injection (Pronk et al, 2007; Haas et al, 2015) than those in WT mice (Fig. 1E,F). After the resolution of 5-FU-induced stress, *Hmga2* KI mice showed similar BM cell counts and HSCs on day 30 after the 5-FU injection to those in WT mice (Fig. 1C,D), suggesting that the overexpression of *Hmga2* drove the rapid recovery of hematopoiesis in response to the 5-FU injection.

To elucidate the mechanisms by which *Hmga2* KI HSCs expanded and produced more progenitor cells soon after the 5-FU injection, we examined post-injection changes in the cell cycle of *Hmga2* KI HSCs. In the pre-injection period, we found a similar cell cycle status between WT and *Hmga2* KI HSCs, which were mostly quiescent in the G0 stage in BM; however, *Hmga2* KI HSCs showed higher frequencies of cells in the S/G2/M stages on day 7 and 12 after the 5-FU injection, while WT HSCs showed transient increases in the S/G2/M stages on day 7, but started to become quiescent 12 days after the injection (Fig. 1G). We next investigated whether *Hmga2* KI HSCs maintained the stem cell phenotype or underwent differentiation after cell division. We stained the plasma membrane of cells using CytoTell and analyzed HSC frequencies after cell division in both in vitro and in vivo settings (Umemoto et al, 2022). In the in vitro culture, *Hmga2* KI HSCs proliferated and maintained more phenotypic stem cells in the second and fourth divisions than WT HSCs (Fig. 1H). Based on previous findings showing that HSCs started to proliferate 4 days after an injection of 5-FU (Umemoto et al, 2022), we transplanted plasma membrane-stained HSCs into 5-FU-treated mice on day 4 and measured the percentage of phenotypic HSCs in BM on day 7 after the 5-FU injection (Fig. 1I). WT HSCs mostly underwent differentiation after the first division, while more than half of *Hmga2* KI HSCs maintained phenotypic HSCs after the first division in BM after the injection of 5-FU (Fig. 1J). Overall, in the steady state, the overexpression of *Hmga2* did not change the quiescent status of HSCs in BM, but expanded HSCs and erythroid/MkP cells under stress conditions, resulting in the faster recovery of hematopoiesis.

### The Hmga2 gene was necessary for the expansion of HSCs and MkPs after the 5-FU injection

Hmga2 was previously shown to be necessary for the high self-renewal capacity of fetal HSCs (Copley et al, 2013). Since the overexpression of *Hmga2* expanded adult HSCs under stress conditions, we employed a loss-of-function of *Hmga2* approach to establish whether the *Hmga2* gene was necessary for adult HSCs and the recovery of hematopoiesis after the 5-FU injection. To achieve this, we generated a new *Hmga2^flox/flox^* cKO mouse of the *Hmga2* gene, in which exons 2 and 3 were fused by deleting 10.7 kb of intron 2, which does not contain potentially transcribed sequences in blood cells, and were surrounded by two loxP sites (Fig. 2A). *Hmga2^flox/flox^* mice showed similar expression levels of the *Hmga2* gene in HSCs to

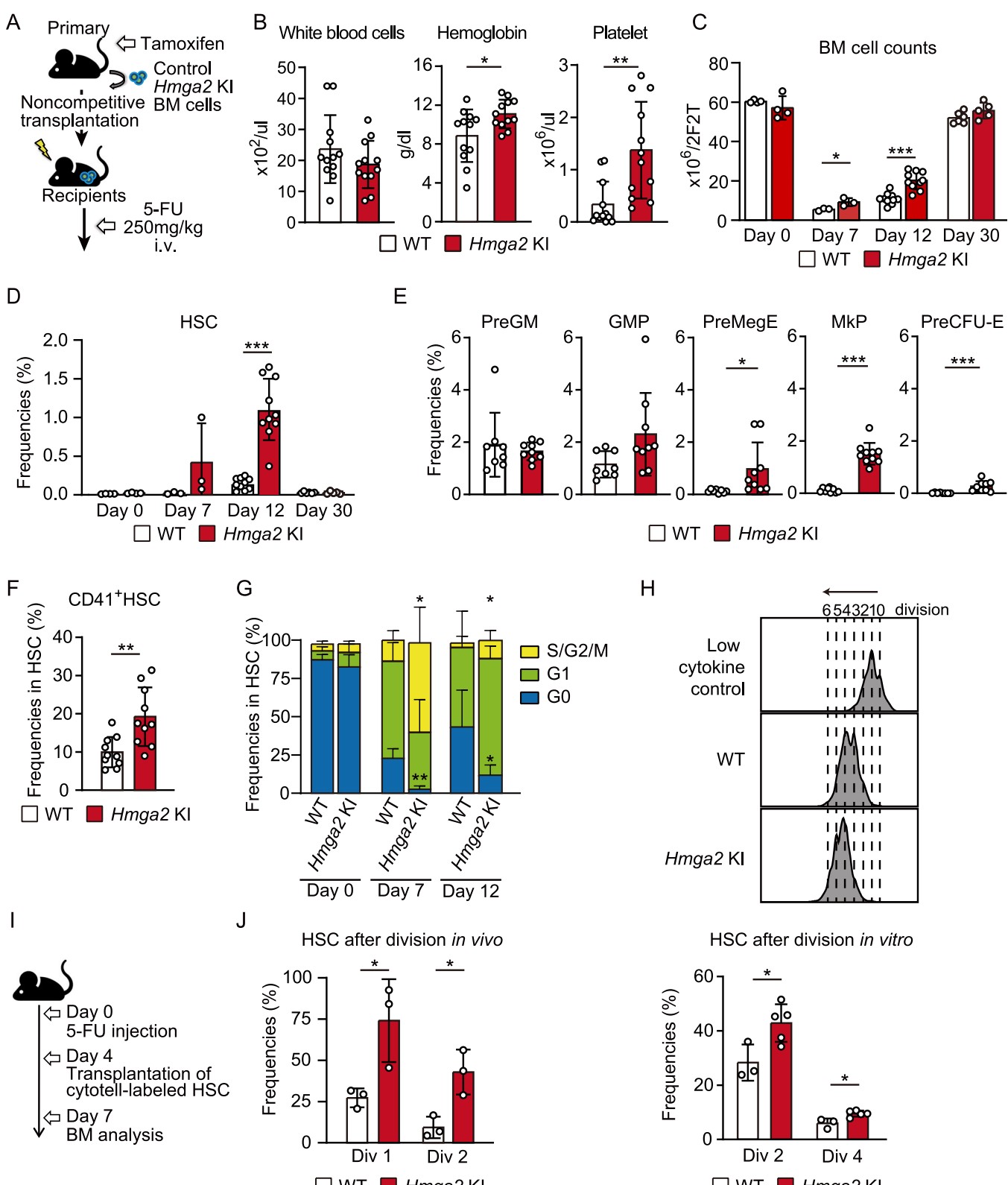

those in primary wild-type mice (Fig. 2B). *Hmga2* cKO mice were crossed with *Cre-ERT2* mice, and these BM cells were transplanted into lethally-irradiated CD45.1⁺ WT recipient mice (Fig. 2C). *Hmga2^{flox/flox}* mice and *Cre-ERT2* mice served as controls. After the

tamoxifen injection, we confirmed that the expression of *Hmga2* mRNA in HSCs was lower in *Hmga2* cKO mice than in control mice, and the *Hmga2* exon was genetically deleted (Fig. 2D; Appendix Fig. S1). In contrast to the resistant phenotypes of *Hmga2* KI mice to

**Figure 1. Hmga2 overexpression expanded HSCs and recovered hematopoiesis after the 5-FU injection.**

(A) Schematic illustration showing primary transplanted mice transplanted with BM MNCs one month after the tamoxifen treatment, injected with 250 mg/kg 5-FU into the tail vein 6 weeks after transplantation, and analyzed 0, 7, 12, and 30 days after the 5-FU injection. (B) Complete blood cell counts (CBC) in the PB of WT and Hmga2 KI mice 12 days after the 5-FU injection ($n = 12$). (C) BM cell counts 0, 7, 12, and 30 days after the 5-FU injection ($n = 3$–10). (D) Percentages of Lin$^-$EPCR$^+$CD48$^-$CD150$^+$ HSCs in the BM of mice 0, 7, 12, and 30 days after the 5-FU injection ($n = 3$–10). (E) Percentages of Pre-GM (Lin$^-$Kit$^+$Sca-1$^-$FcgR$^-$CD41$^-$CD105$^-$CD150$^-$), GMP (Lin$^-$Kit$^+$Sca-1$^-$FcgR$^+$CD41$^-$CD150$^-$), PreMegE (Lin$^-$Kit$^+$Sca-1$^-$FcgR$^-$CD41$^-$CD105$^-$CD150$^+$), MkP (Lin$^-$Kit$^+$Sca-1$^-$CD41$^+$CD150$^+$), and Pre-CFU-E (Lin$^-$Kit$^+$Sca-1$^-$FcgR$^-$CD41$^-$CD105$^+$CD150$^+$) in the BM of mice 12 days after the 5-FU injection ($n = 9$–10). (F) Frequencies of CD41$^+$ cells among HSCs in BM 12 days after the 5-FU injection ($n = 10$). (G) Cell cycle profiles of HSCs in the BM of mice 7 and 12 days after the 5-FU injection ($n = 3$). (H) Frequencies of Lin$^-$Kit$^+$Sca-1$^+$CD48$^-$CD150$^+$ HSCs after the second and fourth divisions in a liquid culture among CytoTell-labeled cells in BM ($n = 3$–5). A culture with low cytokine levels was used as an undivided control. (I) Schematic illustration showing 5-FU-injected wild-type mice transplanted with CytoTell-labeled WT or Hmga2 KI HSCs 4 days after the 5-FU injection, in which BM was analyzed 3 days after transplantation. (J) Frequencies of HSCs after the first and second division among CytoTell-labeled cells in BM ($n = 3$). Data information: In panel (B–H, J), bars show the mean ± SD, *$p < 0.05$, **$p < 0.01$, and ***$p < 0.001$. N means the number of mice in panel (B–G) and the number of samples in panel (H, J). P-values were calculated by the Student's t-test. Data were combined from two, three, or four independent experiments. Source data are available online for this figure.

the 5-FU injection, platelet counts after the injection of 5-FU (Fig. 2E), BM cell counts (Fig. 2F), and the frequencies of HSCs and MkP cells 12 days after the 5-FU injection were lower in Hmga2 cKO mice than in control mice (Fig. 2G,H). No significant changes were observed in hemoglobin levels or erythrocyte progenitor cell counts in BM between control and Hmga2 cKO mice after the 5-FU injection (Fig. 2E,H). The present results showed that 6 out of 10 Hmga2 cKO mice and 2 out of 13 WT mice died 30 days after the 5-FU injection (Fig. 2I); however, surviving Hmga2 KO mice showed similar BM cell counts and HSC numbers to those in WT mice 30 days after the 5-FU injection (Fig. 2F,G). These results suggest that the Hmga2 gene was critical for the expansion of HSCs and the recovery of hematopoiesis soon after the 5-FU injection.

## The overexpression of Hmga2 maintained the expression of stem cell genes and suppressed the up-regulation of inflammatory response genes under stress conditions

We investigate the mechanisms by which the overexpression of Hmga2 regulated the transcriptome in HSCs under homeostatic and stress conditions. We initially performed RNA sequencing on fetal WT HSCs and adult CD150$^+$CD48$^-$CD34$^-$CD135$^-$LSK HSCs and MPP2-4 cells isolated from WT and Hmga2 KI mice two months after transplantation. In comparison with WT HSCs, Hmga2 KI HSCs showed changes in the number of >2-fold differentially expressed genes (DEGs), such as the upregulated expression of Hmga2 and Igf2bp2, a major target gene of Hmga2 in tissue stem cells (Li et al, 2012; Bai et al, 2021) (Dataset EV1). Consistent with similar behaviors in hematopoiesis between WT and Hmga2 KI mice in the steady state (Sun et al, 2022), a principal component analysis (PCA) using the transcriptomes of all genes confirmed that Hmga2 KI HSCs were closer to WT HSCs in the steady state, while both adult HSCs were located far from highly-proliferative fetal HSCs separated by the first principal component (PC) (Fig. 3A). The second PC likely resembled differentiation from adult HSCs to MPPs (Fig. 3A).

To elucidate the mechanisms by which Hmga2 KI mice changed the transcriptome in HSCs to expand HSCs under stress conditions, we performed RNA sequencing on WT and Hmga2 KI Lineage$^-$CD150$^+$CD48$^-$EPCR$^+$ HSCs on days 0, 3 and 6 after the in vivo injection of 5-FU. PCA revealed that WT and Hmga2 KI HSCs on day 0 were in close proximity; however, the first PC separated WT HSCs on days 3 and 6 after the 5-FU injection from other HSCs, showing 5-FU-induced transcriptional changes (Fig. 3B). There were

more DEGs between WT and Hmga2 KI HSCs on day 3 than on day 0 (Fig. 3C). By using DEGs between Hmga2 KI and WT HSCs on day 3 (Fig. 3C; Dataset EV2), a hierarchical clustering map confirmed that WT HSCs on day 3 formed clusters of genes, showing the downregulated expression of stem cell genes (e.g., Gfi1 and Hlf) and upregulated expression of inflammatory pathway genes (e.g., Tlr7, Tlr8, and Ifi204), while Hmga2 KI HSCs on day 3 rarely shared the DEGs identified in WT HSCs on day 3 (Fig. 3D). Q-RT-PCR confirmed that Hmga2 KI HSCs mitigated 5-FU-induced transcriptional changes in these genes (Fig. EV3). A gene ontology (GO) analysis indicated that the genes upregulated in WT HSCs on day 3 were involved in differentiation, infection, and inflammation, which were mitigated in Hmga2 KI HSCs on day 3 (Fig. 3E). A gene set enrichment analysis (GSEA) revealed that Hmga2 KI HSCs on days 3 and 6 significantly activated cell cycle regulators, such as MYC and E2F target genes (Fig. 3F), and repressed genes involved in inflammation, including the TNF-α-NF-kB, interferon-γ, and JAK-STAT3 pathways, more than WT HSCs on days 3 and 6, respectively (Fig. 3F). GSEA also revealed that in comparison with WT HSCs, Hmga2 KI HSCs showed the positive enrichment of stem cell-signature genes, but the repression of genes involved in differentiation (Fig. 3G). These results suggest that the overexpression of Hmga2 maintained the expression of cell-cycle regulators and stem cell-signature genes, but also suppressed the up-regulation of differentiation and inflammatory response genes in HSCs, thereby driving the regeneration of hematopoiesis without the attrition of HSCs after the 5-FU injection.

## The Hmga2 protein was accumulated by inflammatory cytokines and newly bound to genes involved in proliferation and inflammation pathways

Based on the result showing that the overexpression of Hmga2 drove the regeneration of hematopoiesis, we hypothesized that HSCs may increase the expression of Hmga2 to regulate the transcription of target genes under stress conditions. WT and Hmga2 KI Lineage$^-$CD150$^+$CD48$^-$EPCR$^+$ HSCs both showed slightly lower expression levels of Hmga2 mRNA after the 5-FU injection (Fig. 4A); however, Hmga2 protein expression levels were significantly increased in the HSCs of both genotypes 3 days after the 5-FU injection (Fig. 4B), indicating that Hmga2 protein expression was upregulated in a transcription-independent manner. An in vivo treatment with 5-FU has been shown to induce systemic inflammation in BM (Hérault et al, 2017), and we found that 5-FU-treated BM

plasma cells had higher levels of secreted inflammatory cytokines, such as IL-1, IL-6, TNF-α, and IFN-γ, but not IL-3, one day after the 5-FU injection than control BM plasma cells (Fig. 4C), suggesting that these 5-FU-induced cytokines increase the expression of the Hmga2 protein in HSCs. As expected, a treatment with TNF-α or Toll-like receptor ligands, such as lipopolysaccharide and Pam3CSK4, upregulated the expression of the Hmga2 protein in HSCs in a liquid culture, whereas a treatment with murine IL-3 or 5-FU did not (Fig. 4D). Since inflammatory stimuli upregulated the expression of the Hmga2 protein in HSCs under stress conditions, we found that *Hmga2* KI HSCs increased the number of HSCs in a liquid culture supplemented with 100 ng/ml TNF-α, whereas WT HSCs did not (Fig. 4E). In contrast, we found that *Hmga2* cKO cells significantly decreased the number of HSCs after an in vitro treatment with 100 ng/ml TNF-α (Fig. 4F). We then performed an in vivo treatment with TNF-α (100 μg/kg three times in two days) on *Hmga2* KI and *Hmga2* cKO mice (Fig. 4G). Consistent with in vitro results, the number of HSCs after the TNF-α injection was higher in *Hmga2* KI mice than in WT mice (Fig. 4H), but were lower in *Hmga2* cKO mice than in WT mice (Fig. 4I), suggesting the importance of the *Hmga2* gene for the protection of HSCs and also that its overexpression facilitated the expansion of HSCs in response to inflammatory cytokines.

We then investigated whether the Hmga2 protein newly bound to chromatin to regulate the transcription of genes in HSCs after the 5-FU injection. Since there was no available anti-murine Hmga2 antibody for ChIP, we performed ChIP-seq using an anti-HA antibody and HSPCs isolated from *HA*-tagged-*Hmga2* KI mice 6 days after the injection of 5-FU. Hmga2-ChIP-seq peaks revealed the significant enrichment of AT-rich sequences due to the AT-hooks of Hmga2, supporting the accuracy of anti-HA antibody ChIP (Fig. EV4). As expected, the 5-FU injection significantly increased Hmga2-binding regions in cells, most of which were not bound by Hmga2 in cells before the 5-FU injection (738 peaks on day 6 versus 194 peaks on day 0) (Fig. 4J; Dataset EV3). Since Hmga2-ChIP-seq revealed that the Hmga2 protein rarely bound to promoter and enhancer regions marked by histone modifications, such as H3K27ac (Fig. EV4), we annotated these 5-FU-induced Hmga2-binding regions into proximal genes, and assessed changes in the transcription of annotated genes in WT and *Hmga2* KI HSCs on days 3 and 6 after the 5-FU injection. Before the injection, we found similar activated genes among Hmga2-binding genes between WT and *Hmga2* KI HSCs forming cluster 4 (Fig. 4K). On days 3 and 6 after the injection, WT HSCs increased the expression of genes forming cluster 1 involved in inflammatory diseases defined by the GO analysis (Fig. 4L; Dataset EV4), the activation of which was mitigated in *Hmga2* KI HSCs. *Hmga2* KI HSCs increased the expression of genes forming cluster 2 involved in the cell cycle and proliferation pathways (Fig. 4L).

To establish whether the *Hmga2* gene was needed to regulate the transcription of these target genes, we next performed RNA sequencing of WT and *Hmga2* cKO Lineage⁻CD150⁺CD48⁻EPCR⁺ HSCs on days 0 and 6 after the 5-FU injection (Fig. 2C). We identified 230 upregulated genes and 144 downregulated genes in *Hmga2* cKO HSCs, relative to those in WT HSCs 6 days after the injection. Among these genes, *Hmga2* cKO HSCs shared 59 out of 230 upregulated genes with downregulated genes in *Hmga2* KI HSCs (Fig. 4M), suggesting that the Hmga2 protein directly

suppressed the transcription of these genes. The GO analysis revealed that the upregulated genes in *Hmga2* cKO HSCs enriched genes involved in inflammation (Fig. 4N; Dataset EV5). Moreover, GSEA showed that *Hmga2* cKO HSCs on day 6 showed significantly more positive enrichment in the hallmark TNF-α-NF-kB pathway and TNF-α-mediated pro-death genes than WT HSCs on day 6 (Yamashita and Passegué, 2019) (Fig. 4O). Collectively, these results suggest that the Hmga2 protein was accumulated by inflammatory cytokines and appeared to be necessary for resolving the stress-induced activation of inflammatory genes in HSC. Under stress conditions, the Hmga2 protein newly bound to the distal non-enhancer regions of genes involved in proliferation and inflammation, which activated or repressed the transcription of Hmga2 target genes.

## The Hmga2 protein was phosphorylated by CK2 under stress conditions

Based on the result showing that the Hmga2 protein accumulated and newly bound to chromatin under stress conditions, we hypothesized that post-translational modifications to Hmga2 may regulate the chromatin accessibility, transcription, and cellular function of Hmga2 in HSCs. The Hmga2 protein has potential phosphorylation sites in its acidic domain in the C terminus (Fig. 5A), which were phosphorylated by CK2, thereby modulating its DNA-binding capacity in in vitro settings (Sgarra et al, 2009; Shi et al, 2021). CK2 was also shown to phosphorylate proteins downstream of the TNF-α-NF-kB pathway (Wang et al, 2000; Borgo et al, 2021). Although we did not observe changes in the expression levels of CK2 family genes in HSCs after the in vivo injection of 5-FU, we found that the 5-FU injection and in vitro TNF-α treatment both changed the phosphorylation levels of canonical CK2 substrates (Appendix Fig. S2). To identify which proteins were phosphorylated in cells under stress conditions, we performed a phosphoproteomics analysis using lineage⁻ mono-nuclear cells (MNCs) pooled from WT mice three days after the 5-FU injection. Phosphoproteomics revealed that 2654 proteins were phosphorylated after the 5-FU injection and the Hmga2 protein at Ser-104, which is one of the five CK2 target sites on the protein (Fig. 5A), was phosphorylated before and after the 5-FU injection (Dataset EV6). An immunoprecipitation analysis of immature c-Kit⁺ *Hmga2* KI BM cells revealed that the phosphorylation level of the Hmga2 protein was higher after than before the 5-FU injection (Fig. 5B). We then confirmed that the Hmga2 protein (Hmga2-WT) was directly phosphorylated by CK2 in an in vitro setting, and this was diminished by the substitution of alanine at these five sites in the Hmga2 protein (Hmga2-5A mutant) (Fig. 5C), suggesting that CK2 phosphorylated the Hmga2 protein in immature cells, including HSCs, after the 5-FU injection.

We investigated whether CK2 was required for the expansion of HSCs in mice after the 5-FU injection. We injected 250 mg/kg 5-FU into WT and *Hmga2* KI mice followed by six injections of 25 mg/kg CX-4945, a CK2 inhibitor, and examined the number of HSCs 7 days later (Fig. 5D). The number of HSCs after the treatment with CX-4945 was significantly lower in WT and *Hmga2* KI mice injected with 5-FU than in control DMSO-treated mice (Fig. 5E), suggesting that CK2 phosphorylated the Hmga2 protein to promote the expansion of HSCs under stress conditions.

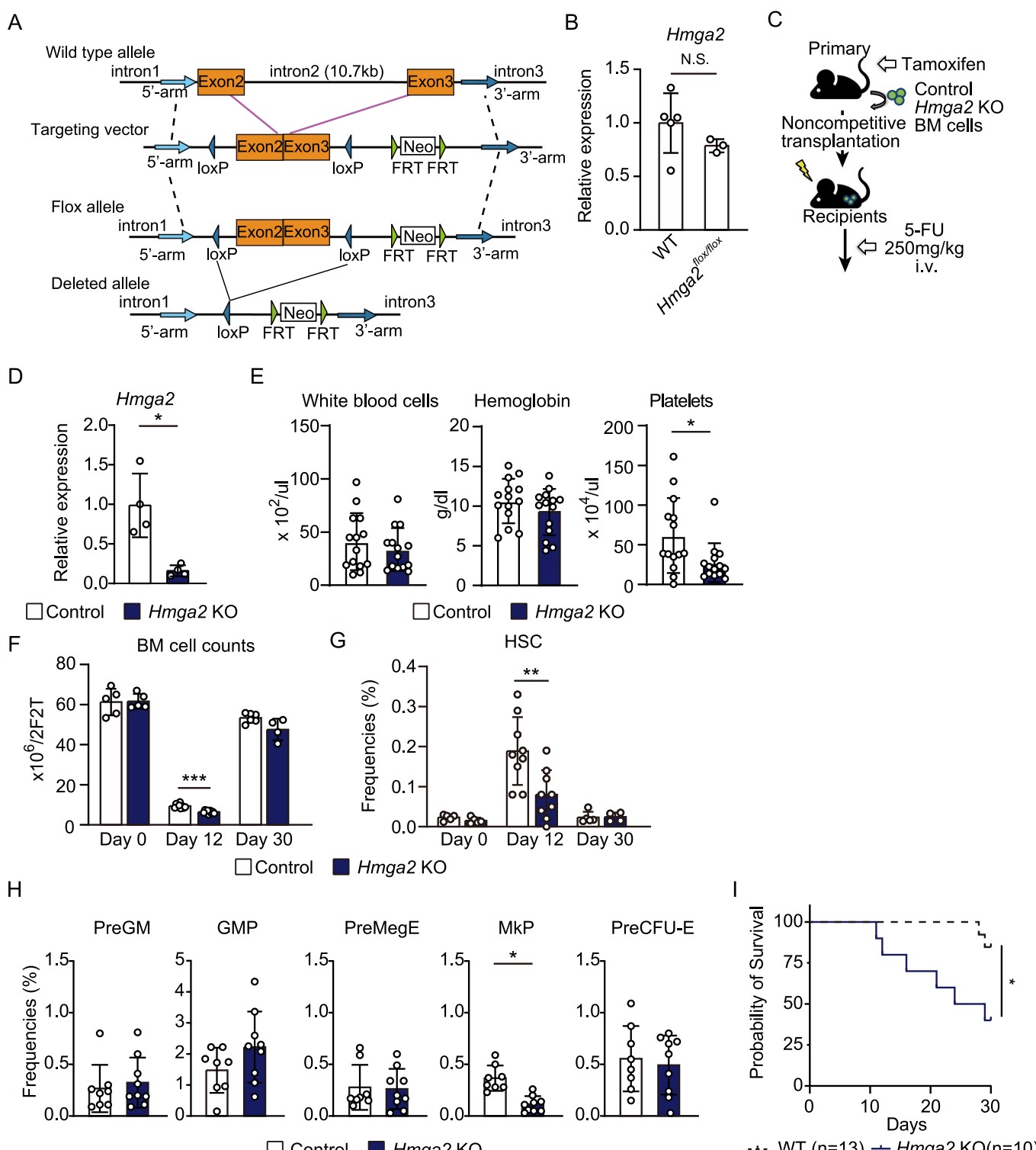

## The TNF-α-CK2-Hmga2 axis regulated the chromatin accessibility, transcription, and expansion of stem cells in response to stress

To clarify the functions of the phosphorylation of Hmga2 for the chromatin binding of Hmga2, we transduced an empty control,

*Hmga2*-WT, or *Hmga2*-5A vector into *Hmga2* cKO HSCs and confirmed similar protein expression levels of WT and mutant Hmga2 in transduced cells (Fig. 6A). After the transplantation of these vector-transduced *Hmga2* KO HSCs into recipient mice (Fig. 6B), Hmga2-ChIP-seq of BM cells revealed that chromatin-binding peaks after the in vitro TNF-α treatment were lower in the

**Figure 2.   Hmga2 was necessary for the expansion of HSCs and megakaryopoiesis after the 5-FU injection.**

(**A**) Generation of a *Hmga2* conditional KO mouse using a targeting vector containing the fusion of exons 2 and 3, both of which code AT-hook domains critical for DNA binding of the Hmga2 protein. (**B**) Expression levels of *Hmga2* in Lin⁻EPCR⁺CD48⁻CD150⁺ HSCs isolated from wild-type mice or *Hmga2^flox/flox* mice ($n = 3-5$). (**C**) Schematic illustration showing primary transplanted mice transplanted with BM MNCs one month after the tamoxifen treatment, injected with 250 mg/kg 5-FU into the tail vein 6 weeks after transplantation, and analyzed 0, 12, and 30 days after the 5-FU injection. (**D**) Expression levels of *Hmga2* in Lin⁻EPCR⁺CD48⁻CD150⁺ HSCs isolated from control mice or *Hmga2^flox/flox;Cre-ERT2* mice after the tamoxifen injection ($n = 4$). Data are representative of two independent experiments. (**E**) CBC of control and *Hmga2* KO mice 12 days after the 5-FU injection ($n = 14$). (**F**) BM cell counts of control and *Hmga2* KO mice 0, 12, and 30 days after the 5-FU injection ($n = 4-9$). (**G**) Percentages of Lin⁻EPCR⁺CD48⁻CD150⁺ HSCs in mice 0, 12, and 30 days after the 5-FU injection ($n = 4-9$). (**H**) Percentages of Pre-GM, GMP, PreMegE, MkP, and Pre-CFU-E in the BM of mice 12 days after the 5-FU injection ($n = 9$). (**I**) Survival of WT and *Hmga2* cKO mice after the 5-FU injection ($n = 10-13$). *P*-values were calculated by the Log-rank test, $^*p < 0.05$. Data were combined from three independent experiments. Data information: In panel (**B, D–H**), bars show the mean ± SD, $^*p < 0.05$, $^{**}p < 0.01$, and $^{***}p < 0.001$. *N* means the number of samples in panel (**B, D**), and the number of mice in panel (**E–I**). *P*-values were calculated by the Student's *t*-test. Data were combined from two, three, or four independent experiments. Source data are available online for this figure.

Hmga2-5A mutant than in Hmga2-WT (Fig. 6C). Since Hmga2 was shown to bind to and open chromatin (Zhao et al, 1993; Kishi et al, 2012), we assessed regional chromatin accessibility driven by Hmga2 in cultured HSCs treated with 100 ng/ml TNF-α for 24 h by ATAC using sequencing (ATAC-seq) (Fig. 6D). A hierarchical clustering map showed that Hmga2-WT-transduced HSCs shared increased and decreased ATAC peaks after the treatment with TNF-α forming clusters 1 and 3, respectively, with both empty control- and Hmga2-5A-transduced HSCs (Fig. 6E). In comparison with Hmga2-WT-transduced HSCs, Hmga2-5A-transduced HSCs enhanced and reduced ATAC peaks after the TNF-α treatment forming clusters 2 and 5, respectively (Fig. 6E).

To investigate what TFs bound to chromatin in HSCs due to the phosphorylation of Hmga2, we performed ATAC-seq TF footprints analysis (Bentsen et al, 2020). As expected, the NF-kB binding motif was enhanced more in TNF-α-treated HSCs than in PBS-treated HSCs (Fig. EV5; Dataset EV7). After the treatment with TNF-α, the motifs of GATA family genes were enhanced, while those of AP-1 TFs were reduced in Hmga2-WT- and Hmga2-5A-transduced HSCs relative to those in empty control-transduced HSCs (Fig. EV5). Notably, in comparison with Hmga2-5A-transduced HSCs, the motifs of CEBP family genes were enhanced and those of RFX family genes were reduced in Hmga2-WT-transduced HSCs (Fig. 6F), indicating that the phosphorylation of Hmga2 was critical for regulating chromatin accessibility and modulating the chromatin binding of TFs in response to the TNF-α treatment.

Based on the results showing that Hmga2 induced open or closed chromatin in HSCs in a phosphorylation-dependent manner, we performed RNA sequencing on *Hmga2*-5A-expressing HSCs after the treatment with TNF-α (Fig. 6D). The transcriptome analysis revealed that among the TNF-α-induced Hmga2-WT-target genes identified by ChIP-seq (Fig. 6C), Hmga2-WT and Hmga2-5A both mildly activated the transcription of genes involved in infection and ribosome pathways to lesser degrees than the empty control in HSCs after the treatment with TNF-α, forming cluster 5 (Fig. 6G; Dataset EV8). In addition, GSEA revealed that the expression levels of TNF-α-mediated pro-death genes were lower in Hmga2-WT-transduced HSCs than in Hmga2-5A-transduced HSCs after the treatment with TNF-α, whereas those of TNF-α-mediated survival genes were similar (Yamashita and Passegué, 2019) (Fig. 6H). Among the RFX family, the Rfx5 gene was shown to activate the expression of MHC class II genes (Clausen et al, 1998), which were expressed in tissue stem cells to maintain the integrity of stem cells and inhibit their transformation

(Beyaz et al, 2021; Hernández-Malmierca et al, 2022). Based on the result showing that Hmga2 reduced the footprint of the Rfx5 TF in a phosphorylation-dependent manner (Fig. 6F), Hmga2-WT-transduced HSCs more strongly repressed the expression of genes in MHC class II antigen presentation than Hmga2-5A-transduced HSCs (Fig. 6H), suggesting that the phosphorylation of Hmga2 drove transcriptional programs to maintain HSCs under stress conditions.

To clarify whether the phosphorylation of Hmga2 enhanced the expansion of HSCs under stress conditions, we examined the growth capacity of these vector-transduced HSCs in a liquid culture supplemented with TNF-α. We found that the cell counts of phenotypic HSCs after the in vitro TNF-α treatment were higher in Hmga2-WT-transduced HSCs than in the other groups (Fig. 6I). We also performed a competitive transplantation assay using these vector-transduced *Hmga2* KO HSCs and control WT HSCs (Fig. 6B), and found that the competitive repopulating capacity of Hmga2-WT-transduced HSCs was significantly higher than that of Hmga2-5A-transduced *Hmga2* KO HSCs 4 months after transplantation (Fig. 6J). We then investigated whether the ectopic expression of *RFX* family genes, such as *Rfx3* and *Rfx5*, inhibited the expansion of *Hmga2* KI HSCs in a liquid culture supplemented with TNF-α (Fig. 6K). The results obtained showed that the transduction of *Rfx5* reduced the number of WT and *Hmga2* KI HSCs significantly more than that of empty control-transduced HSCs (Fig. 6L), suggesting that the phosphorylation of Hmga2 promoted the expansion of HSCs, presumably by inhibiting the binding of the Rfx5 TF to chromatin. Collectively, these results suggest that the TNF-α-CK2-Hmga2 axis regulated chromatin accessibility, the chromatin binding of TFs, and the transcription of target genes, which were integrated to drive the growth and survival of HSCs in response to stress.

## Stress-induced Hmga2-mediated gene signatures were activated in human myelodysplastic syndrome (MDS) HSPCs

The overexpression of *Hmga2* induced a myeloproliferative phenotype in a transgenic mouse (Ikeda et al, 2011), and drove the development of myeloproliferative neoplasms induced by the *JAK2^{V617F}* mutation or *Tet2* deletion in mouse models (Shimizu et al, 2016; Sashida et al, 2016; Bai et al, 2021). An analysis of published datasets on normal and MDS cells in humans revealed that *HMGA2* mRNA expression levels were lower in normal aged HSCs than in young HSCs, and were higher in CD34⁺ HSPCs in

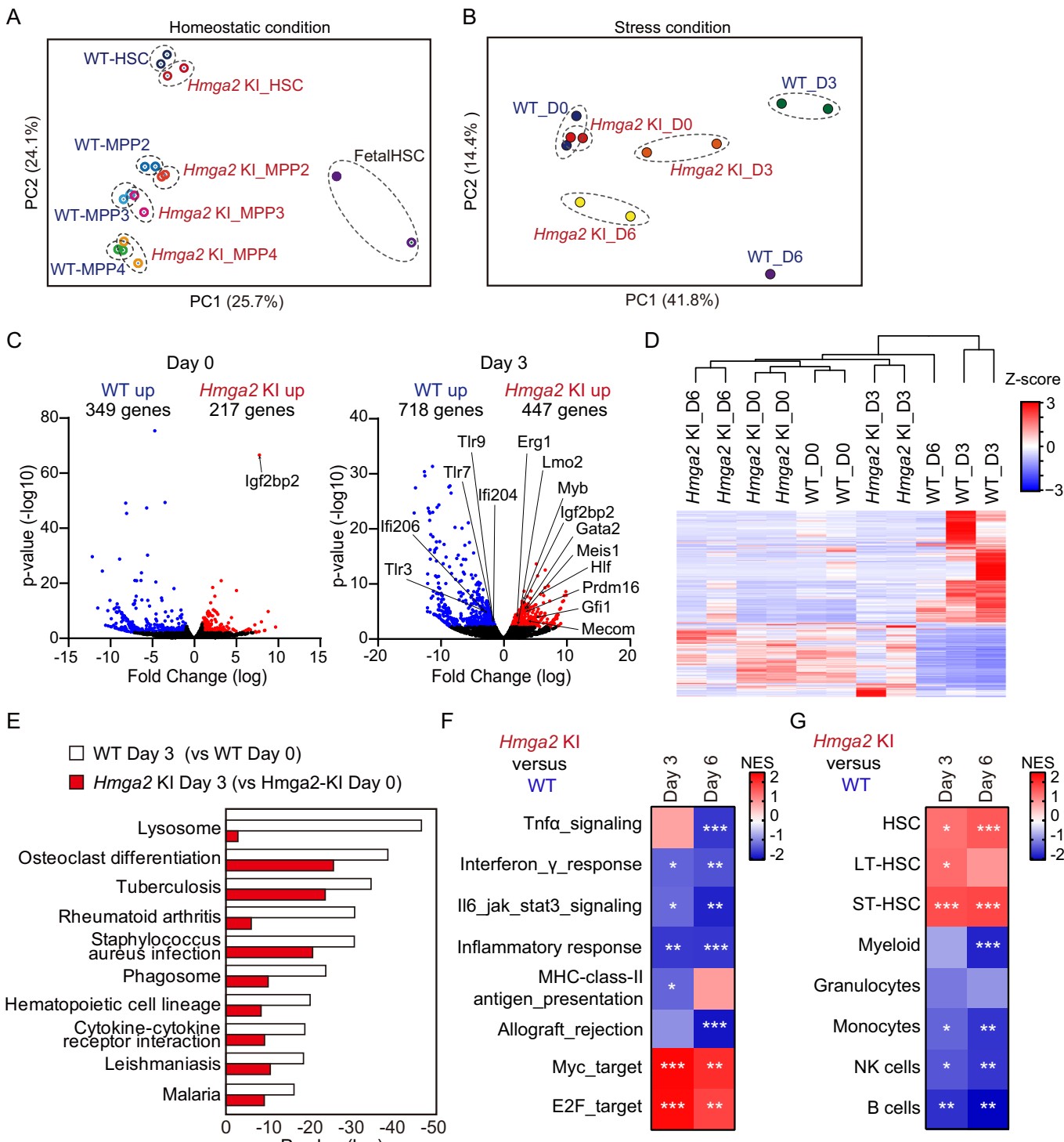

MDS patients than in healthy individuals (Appendix Fig. S3), suggesting that the *HMGA2* gene was activated at least at the mRNA level during the pathogenesis of MDS. Since TNF-α was shown to be increased in the BM of human with clonal hematopoiesis and MDS (Cook et al, 2019; Sawanobori et al, 2003; Stifter et al, 2005), and the activation of TNF-α signaling

pathways in HSPCs drove the development of MDS (Sallman and List, 2019), we used the 5-FU injection-induced gene signatures identified in *Hmga2* KI HSCs (Fig. 3C) to perform GSEA using these datasets of human MDS and AML (Mills et al, 2009; Pellagatti et al, 2010, 2018; Moison et al, 2022). GSEA revealed the positive enrichment of upregulated genes and negative enrichment of

**Figure 3. Hmga2 overexpression mitigated 5-FU-induced transcriptional changes in HSCs.**

(A) A principal component analysis (PCA) based on the expression of all genes in pooled HSCs and MPP1-4 isolated from WT and *Hmga2* KI mice and those in fetal HSCs (linked in DRA013143) (*n* = 2). (B) PCA based on the expression of all genes in WT and *Hmga2* KI Lin⁻EPCR⁺CD48⁻CD150⁺ HSCs 3 and 6 days after the 5-FU injection (*n* = 1–2). (C) Volcano plot showing differentially expressed genes in *Hmga2* KI HSCs from those in WT HSCs (*n* = 2) before and 3 days after the 5-FU injection (*n* = 2). Upregulated genes are shown in red (*p* < 0.01, log2-fold change ≥ 1) and downregulated genes are shown in blue (*p* < 0.01, log2-fold change ≤ −1). (D) Hierarchical clustering based on differentially expressed genes between WT and *Hmga2* KI HSCs 3 days after the 5-FU injection (*n* = 1–2). (E) GO analysis of differentially expressed genes in WT and *Hmga2* KI HSCs 3 days after the 5-FU injection from those before the injection (*n* = 2). (F, G) GSEA of the hallmark gene sets, the C2 MHC class II antigen presentation and hematopoietic lineage-specific signatures (linked in GSE1559 and GSE6506) in *Hmga2* KI HSCs 3 and 6 days after the 5-FU injection relative to those in WT HSCs (*n* = 2). Asterisks show *FDR < 0.25, **FDR < 0.05, and ***FDR < 0.01 with *p* < 0.05. Data information: *N* means the number of samples in panel (A–G). In panel (C), fold changes and *p*-value were calculated by EDGE software (3.38.4). In panel (E), *p*-values were calculated by using gene ontology functions of HOMER software. Source data are available online for this figure.

downregulated genes in human MDS HSPCs, irrespective of genetic mutations, relative to those in control HSPCs (Fig. 7A). This result suggests that the overexpression of *HMGA2* functioned in MDS HSPCs partly through stress responses.

## Discussion

We demonstrated that Hmga2 was vitalized by the TNF-α-CK2 signaling for the expansion of HSCs to conduct stress hematopoiesis. Hmga2 are involved in regulation of chromatin accessibility and the transcriptome downstream of the TNF-α-CK2 axis in HSCs. Under stress conditions, such as transplantation and after the 5-FU injection, which elevated inflammatory cytokines in BM plasma cells *Hmga2* KI mice showed the enhanced expansion of HSCs and rapid hematopoietic recovery increasing megakaryocyte-biased HSCs and MkP cells in BM and platelets in PB. In contrast, after the 5-FU injection, *Hmga2* cKO mice decreased the number of HSCs and showed delayed hematopoietic recovery, which resulted in the death of *Hmga2* cKO mice, suggesting that the presence of *Hmga2* was critical for stress hematopoiesis. In fact, adult HSCs increased the expression of the endogenous Hmga2 at the protein, but not mRNA level in response to stress stimuli, including TNF-α, which induced the phosphorylation of the Hmga2 protein at its acidic domain due to the activation of CK2 kinase, thereby forming the TNF-α-CK2-Hmga2 axis. In contrast, under homeostatic conditions, neither *Hmga2* cKO nor *Hmga2* KI mice showed a significant change in hematopoietic phenotypes, indicating that the *Hmga2* gene was not necessary for normal steady-state hematopoiesis. The present results revealed a role for the TNF-α-CK2-Hmga2 axis in the expansion and survival of HSCs and the regeneration of hematopoiesis driving megakaryopoiesis under stress conditions (a model is shown in Fig. 7B).

A stimulation with TNF-α was considered to be detrimental for HSCs (Pronk et al, 2011; Ishida et al, 2017), whereas the TNF-α-mediated activation of NF-kB was recently shown to prevent HSC necroptosis (Yamashita and Passegué, 2019). In development, sterile TNF-α signaling is required for the generation of HSCs (Espín-Palazón et al, 2014). Therefore, the mechanisms by which a stimulation with TNF-α regulates the transcription of downstream genes and the behavior of HSCs in a context-dependent manner remain unclear. The present study revealed that under stress conditions, including a TNF-α stimulation, the expression of *Hmga2* increased the expansion of HSCs, but not MPP cells, and drove megakaryopoiesis. In this process, the Hmga2 protein was phosphorylated by CK2 downstream of TNF-α, thereby regulating

the chromatin accessibility and chromatin binding of TFs, demonstrating critical roles for the TNF-α-CK2-Hmga2 axis in stress hematopoiesis. NF-kB TF has been implicated in the activation of *Igf2bp2/Imp2* transcription in mouse embryonic fibroblast cells in cooperation with Hmga2 (Cleynen et al, 2007), suggesting a link between these genes and transcriptional regulation in HSCs mediated by TNF-α. The overexpression of *Hmga2* in WT adult HSCs is not sufficient for the development of malignancy in mice and non-human primates (Kumar et al, 2019; Bonner et al, 2021; Sun et al, 2022) and also presumably in humans after gene therapies (Cavazzana-Calvo et al, 2010). Since TNF-α levels are constitutively elevated in aged humans and patients with MDS (Sawanobori et al, 2003; Vasto et al, 2007), the TNF-α-CK2-Hmga2 axis may increase the expression level of *HMGA2* and contribute to the progression of MDS HSCs that harbor driver mutations. Further studies are needed to clarify whether Hmga2 and NF-kB are integrated to regulate transcription for the expansion and survival of normal and malignant HSCs under stress conditions.

CK2 is a critical kinase that regulates various cellular functions in embryos and adult tissues, and its enhanced activation may impair the functions of tissues and lead to cancer development (Borgo et al, 2021). While the AT-hooks of Hmga2 are required for DNA-binding and transcription-activating properties, the phosphorylation of Hmga2 at the C terminus was shown to be mediated by CK2 and affected its DNA-binding capacity in in vitro settings (Sgarra et al, 2009; Maurizio et al, 2011). We consistently confirmed that the acidic domain in the Hmga2 protein was directly phosphorylated by CK2. Furthermore, the phosphorylation of the Hmga2 protein was involved in TNF-α-induced chromatin accessibility to close the binding sites of the Rfx5 TF, which is expressed in HSCs and immune cells, and was also shown to regulate the expression of MHC class II genes (Clausen et al, 1998). The transduction of *Rfx5* decreased the numbers of WT and *Hmga2* KI HSCs after the stimulation with TNF-α, suggesting that the suppression of Rfx5 function was critical for the survival of HSCs under stress conditions. In addition to the phosphorylation-dependent property of inhibiting the binding of Rfx5, under stress conditions, Hmga2 opened the binding sites of GATA family genes and closed those of AP-1 family genes in HSCs in a manner that dwas not dependent on the phosphorylation of Hmga2, and also regulated the transcription of their target genes, which may contribute to the expansion of HSCs and promotion of megakaryopoiesis (Fig. 7B). Although the Hmga2 protein has been proposed to facilitate the chromatin binding of other TFs through their direct association (Fusco and Fedele, 2007), Hmga2-ChIP-seq revealed enriched AT-rich sequences via the

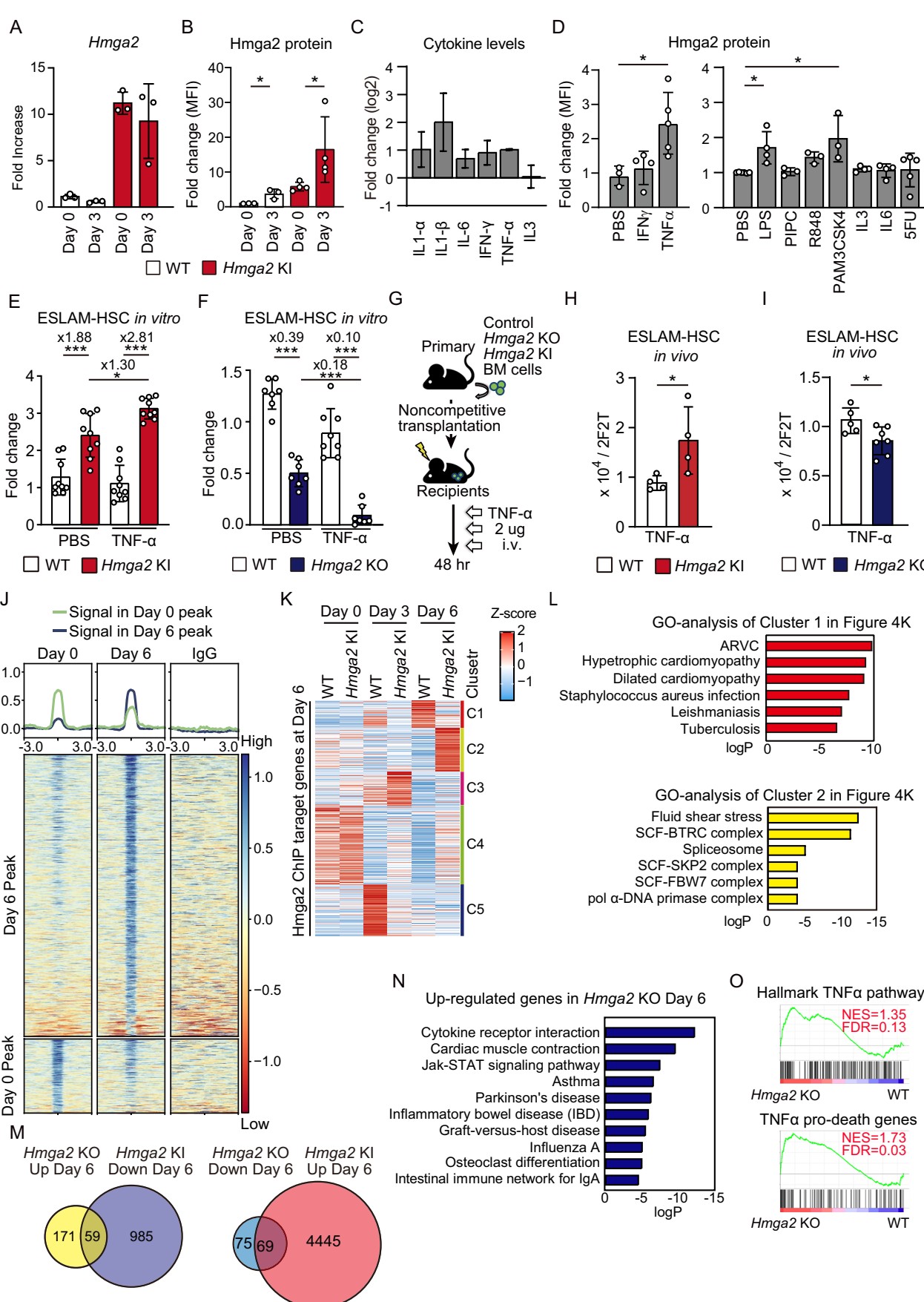

**Figure 4. The TNF-α stimulation increased Hmga2 protein levels and modulated the transcription of Hmga2-binding genes.**

(**A**) *Hmga2* mRNA expression levels in HSCs quantified by q-PCR 3 days after the 5-FU injection ($n = 3$). (**B**) Hmga2 protein expression levels in HSCs 3 days after the 5-FU injection measured by flow cytometry ($n = 3$–4). (**C**) Expression levels of cytokines in BM one day after the 5-FU injection ($n = 2$). (**D**) Hmga2 protein levels in HSCs 24 h after the stimulation with the indicated cytokines ($n = 3$–5). (**E**) Fold changes in the numbers of WT and *Hmga2* KI HSCs in a culture supplemented with 100 ng/ml TNF-α for 3 days relative to those of PBS-treated WT HSCs ($n = 9$). Cultures were started from 500 HSCs. (**F**) Fold changes in the numbers of WT and *Hmga2* KO HSCs in a culture supplemented with 100 ng/ml TNF-α for 3 days relative to those of PBS-treated WT HSCs ($n = 7$–8). Cultures were started from 1000 HSCs. (**G**) Schematic illustration showing primary transplanted mice transplanted with BM MNCs one month after the tamoxifen treatment, injected with 100 µg/kg of TNF-α three times every 12 h into the tail vein 6 weeks after transplantation, and from which BM was analyzed two days after the first TNF-α injection. (**H, I**) Numbers of Lin⁻EPCR⁺CD48⁻CD150⁺ HSCs in BM in WT, *Hmga2* KI, and *Hmga2* KO mice two days after the first TNF-α injection ($n = 4$–7). (**J**) Heatmaps showing the levels of Hmga2-binding peaks in pooled *Hmga2* KI HSPCs isolated from mice 6 days after the 5-FU injection. Data are representative of two independent experiments. (**K**) Heatmap showing the z-scores of expression values in 5-FU-induced Hmga2-binding genes in the indicated HSCs, forming five clusters based on K-means clustering. (**L**) GO analysis of differentially expressed genes in clusters 1 and 2 detected in Fig. 4K. (**M**) Venn diagrams of up- and down-regulated genes in pooled *Hmga2* KO HSCs and *Hmga2* KI HSCs relative to those in WT HSCs 6 days after the 5-FU injection ($n = 2$–4). (**N**) GO analysis of upregulated (>2 fold) genes in *Hmga2* KO HSCs relative to those in WT HSCs 6 days after the 5-FU injection ($n = 4$). (**O**) GSEA of the hallmark of TNF-α-NF-kB signaling and TNF-α-dependent pro-death genes (linked GSE115403) in *Hmga2* KO HSCs 6 days after the 5-FU injection relative to those in WT HSCs ($n = 4$). Data information: In panel (**A–F, H, I**), bars show the mean ± SD, *$p < 0.05$, **$p < 0.01$, and ***$p < 0.001$. *P*-values were calculated by the Student's *t*-test. In panel (**L, N**), *p*-values were calculated by using gene ontology functions of HOMER software. *N* means the number of samples in panel (**A–F, M–O**), and the number of mice in panel (**H, I**). Data were combined from two or three independent experiments. Source data are available online for this figure.

AT-hook of Hmga2, but no enriched TF-binding motifs in the Hmga2-binding regions in blood cells, which were mostly non-regulatory regions of genes lacking active histone modifications, such as H3K27ac and H3K4me1. Further studies are needed to elucidate the molecular mechanisms by which the C-terminal phosphorylation of Hmga2 modulates its chromatin binding to non-regulatory regions and simultaneously regulates the transcription of target genes mediated by TFs. Nevertheless, the present results suggest critical roles for the acidic domain and its phosphorylation in Hmga2 in the regulation of chromatin accessibility to modulate the transcription for HSCs in stress conditions.

The present study has an important limitation. Our mouse experiments were mostly performed under transplantation and tamoxifen injection settings, which potentially cause severe stress for HSCs, such as increased inflammatory environments in BM and the homing and replication of HSCs. We noted that after the 5-FU injection, the recoveries of progenitor cells and platelets, but not HSCs, were slower in transplanted mice following the tamoxifen injection than in primary mice. Although we compared data from our mouse experiments between WT and *Hmga2* mutant mice subjected to the same transplantation conditions, these manipulations may mask and/or exacerbate the property of Hmga2 and the functionality of HSCs, which may affect our interpretation of the results obtained. Further studies are needed to clarify the role of Hmga2 in HSCs under natural conditions without transplantation.

We demonstrated that Hmga2 dynamically regulated chromatin and the transcription of genes in HSCs accompanied with the phosphorylation of the Hmga2 protein, which affects the fate of HSCs under stress conditions. The present results provide a possibility for therapeutic interventions based on the TNF-α-CK2-Hmga2 axis to promote the regeneration of normal hematopoiesis, but also prevent the expansion of MDS stem cells in patients.

## Methods

### Mice

All mice were of the C57BL/6 background. *Rosa26-flox-stop-flox-HA-Hmga2-eGFP* conditional KI mice were generated as previously reported (Sun et al, 2022), and were crossed with *Rosa26-Cre-ERT2* KI mice (TaconicArtemis). *Hmga2^{flox/flox}* cKO mice were generated using murine C57BL/6 ES cells and the *Hmga2* Exon 2 and Exon 3-fused mini-gene with loxP sequences ligated into the pFFRT-PGKneo vector. *Hmga2^{flox/flox}* KO mice were crossed with *Cre-ERT2* mice and *Vav-iCre* mice. C57BL/6 mice congenic for the *Ly5* locus (CD45.1) were purchased from Sankyo-Lab Service. Adult male or female mice (8–12 weeks) were used as donors and recipients for experiments. Two milligrams of tamoxifen (T5648, Sigma-Aldrich) was administered via an intraperitoneal injection for 5 consecutive days to overexpress and deletion of *Hmga2*. All experiments using these mice were performed in accordance with our institutional guidelines for the use of laboratory animals and approved by the Review Board for Animal Experiments of Kumamoto University (Kumamoto, Japan). Reference numbers were A2020-050 and A2022-022. All mouse experiments were performed without randomization and blinding.

### Cells

293T and 293GPG cell lines were cultured in DMEM containing 10% fetal bovine serum (Ory et al, 1996). Mycoplasma contamination was tested in all cell lines by performing PCR.

### Flow cytometry and antibodies

Flow cytometry and cell sorting were performed using the following anti-murine antibodies purchased from BioLegend or eBioscience (clone and catalog numbers): CD45.2 (104, 109820), CD45.1 (A20, 110730), Gr-1 (RB6-8C5, 108404), Mac1 (M1/70, 101208), Ter119 (116204), IL-7Rα (A7R34, 121104), B220 (RA3-6B2, 103212), CD3e (145-2C11, 100304), CD4 (L3T4, 100526), CD8α (53-6.7, 100714), c-Kit (2B8, 105812), Sca-1 (D7, 108114), FcγRII-III (93, 101308), CD34 (MEC14.7, 11-0341-85), CD41 (MWReg3, 133928), CD48 (HM48-1, 103443), CD105 (MJ7/18, 120414), CD135 (A2F10, 135306), CD150 (TC15-12F12.2, 115924), and CD201 (eBio1560, 2071517). The lineage mixture solution contained biotin-conjugated anti-Gr-1, B220, CD3e, CD8α, Ter119, and IL-7Rα antibodies. All analyses and cell sorting were performed as described in our previous study (Yokomizo-Nakano et al, 2023).

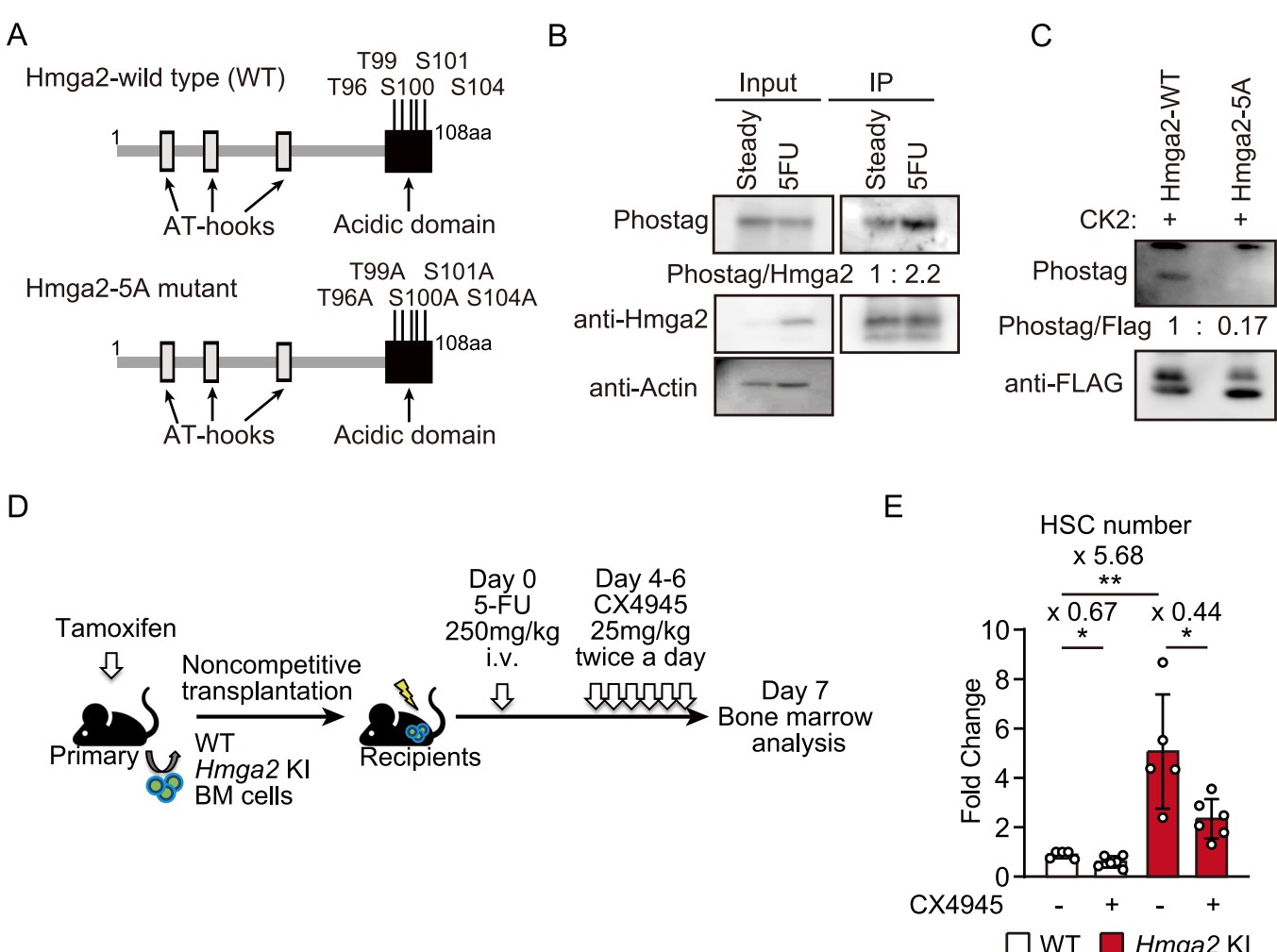

**Figure 5.  The Hmga2 protein was phosphorylated by Casein kinase 2 after the 5-FU injection.**

(A) The Hmga2 protein and five potential phosphorylation sites in the acidic domain. (B) The Hmga2 protein was phosphorylated in MNCs in *Hmga2* KI mice before and 3 days after the 5-FU injection, detected using the Phos-tag antibody on immunoprecipitated Hmga2. (C) CK2 induced the phosphorylation of the Hmga2 protein under in vitro conditions. (D) Schematic illustration showing transplanted mice injected with 250 mg/kg 5-FU, followed by an injection with DMSO or 25 mg/kg CX4945 six times every 12 h, and then analyzed 7 days after the 5-FU injection. (E) Fold changes in the numbers of WT and *Hmga2* KI HSCs after the CX4945 treatment from those in DMSO-treated WT HSCs ($n = 5$–6; $n$ means the number of mice). Bars show the mean ± SD, $*p < 0.05$, and $**p < 0.01$. P-values were calculated by the Student's $t$-test. Data were combined from three independent experiments. (B, C) Data are representative of two or three independent experiments. Source data are available online for this figure.

## Virus vectors and virus transduction

Mouse *Rfx5, Rfx3, Hmga2*, and *Hmga2*-5A lacking 3′UTR were subcloned into the pGCDN-sam-ires-NGFR vector. A VSV-G pseudo-type retroviral supernatant was prepared by transfecting a vector plasmid into 293T cells or 293GPG packaging cells. Virus transduction was performed as described in our previous study (Yokomizo-Nakano et al, 2020). Briefly, purified HSPCs were incubated in serum-free SF-03 medium supplemented with 0.1% BSA, 50 μM 2-ME, 10 ng/mL mouse SCF, and 100 ng/mL mouse TPO for 24 h and then transduced with the indicated virus vectors in the presence of 1 μg/mL RetroNectin and 10 μg/mL protamine sulfate.

## Liquid culture assay

One day after virus transduction, the liquid culture assay was performed after the infection medium was replaced with SF-03 medium supplemented with 0.1% BSA, 50 μM 2-ME, 10 ng/ml mouse SCF, and 100 ng/ml murine TPO. Cells were treated with 100 ng/ml TNF-α (SBI). All in vitro experiments were conducted without randomization or blinding.

## In vitro kinase assay

An in vitro kinase assay was performed as previously reported (Kubota et al, 2013). Cell lysates expressing FLAG-Hmga2 or

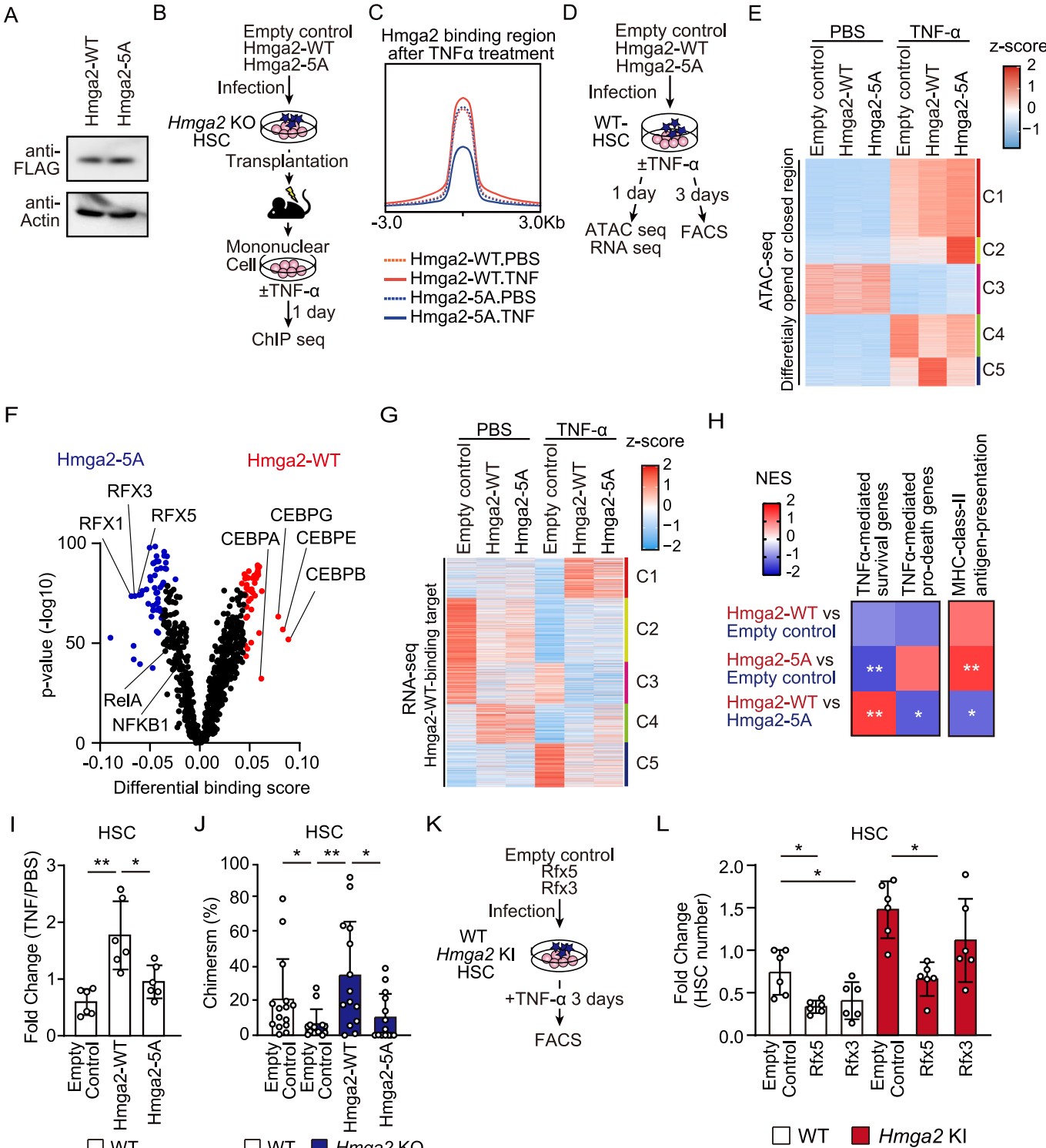

FLAG-Hmga2-5A were immunoprecipitated by anti-FLAG antibody-precoated protein G beads and then incubated with CK2 (NEB) in protein kinase buffer containing 100 M unlabeled ATP at 30 °C for 30 min. Phosphorylated bands were immunoblotted using Phos-tag (Fujifilm).

## CytoTell labeling assay

The CytoTell labeling assay was performed as previously reported (Umemoto et al, 2022). Briefly, purified stem cells were stained with CytoTellTM Red 650 (AAT Bioquest) diluted at 1/500 in PBS for

Figure 6.   The TNF-α-CK2-Hmga2 axis regulated the chromatin accessibility, transcription, and expansion of stem cells in response to stress.

(A) Protein expression levels of Hmga2-WT and the Hmga2-5A mutant in transduced cells. Data are representative of two independent experiments. (B) Schematic illustration of the transduction of the described vectors in *Hmga2* KO HSCs and transplantation, followed by the TNF-α treatment on MNCs in a liquid culture for 24 h. (C) Histogram showing the normalized values of the Hmga2-WT and Hmga2-5A mutant binding peaks in cells with or without the TNF-α treatment ($n = 2$). (D) Schematic illustration of the transduction of the described vectors into WT HSCs and subsequent experiments performed in Fig. 6E–I. (E) Heatmap showing the levels of ATAC peaks in the described HSCs among maximumly 10,000 differentially accessible regions forming five clusters based on K-means clustering ($n = 2$). Two independent experiments data were combined. (F) Volcano plots showing the differential binding activity of all investigated transcription factors between the Hmga2-WT and Hmga2-5A-transduced HSCs after the TNF-α treatment ($n = 2$). (G) Heatmap showing the z-scores of the expression values of TNF-α-induced Hmga2-WT binding genes, assessed in Fig. 6C, in HSCs forming five clusters based on K-means clustering ($n = 3$). (H) GSEA of TNF-α-inducible survival and pro-death genes, which were identified in HSC and GMP, respectively (linked in GSE115403), and the C2 MHC class II antigen presentation in TNF-α-treated described-genotype HSCs ($n = 3$). Asterisks show *FDR < 0.25 and **FDR < 0.05. (I) Fold changes in the numbers of TNF-α-treated HSCs relative to those of PBS-treated HSCs ($n = 3$–6). (J) Frequencies of CD45.2⁺NGFR⁺ cells among HSCs in BM 4 months after transplantation ($n = 13$–15). (K) Schematic illustration of the transduction of empty control, *Rf3*, and *Rfx5* vectors in WT or *Hmga2* KI HSCs, followed by the TNF-α treatment of HSCs in a liquid culture for 3 days. (L) Fold changes in the numbers of the described HSCs relative to those of empty vector-transduced WT HSCs in a culture supplemented with 100 ng/ml TNF-α for 3 days ($n = 6$). Data information: In panel (F), Differentially binding score, transcription factor, and *p*-values were calculated by using TOBIAS software. In panels (I, J, L), bars show the mean ± SD, *$p < 0.05$ and **$p < 0.01$. *P*-values were calculated by the Student's *t*-test. *N* means the number of samples in panel (C, E–I, L), and the number of mice in panel (J). Data were combined from two or three independent experiments. Source data are available online for this figure.

10 min. Stained cells were cultured or transplanted into 5-FU-treated recipient mice and then analyzed 2.5 days after transplantation.

## Cytokine array

Cytokine levels in BM were measured by the Proteome Profiler Mouse Cytokine Array Kit, Panel A (ARY006, R and D Systems). BM was flushed out by 500 μl of PBS and centrifuged at 2000 × *g* for 15 min. Supernatants were applied to a cytokine array.

## Assay for transposase-accessible chromatin using sequencing (ATAC-seq)

ATAC-seq libraries were prepared using the Omni-ATAC protocol (Corces et al, 2017). Briefly, cells were lysed and then incubated with transposase solution containing 1 μL of Tagment DNA Enzyme (Illumina). Transposed DNA was purified and then amplified using a NEBNext High Fidelity 2× PCR Master Mix (New England Biolabs) with indexed primers. The prepared libraries were sequenced on NextSeq500 (Illumina) with 38-bp paired-end reads. Bowtie2 (v2.2.6) was used to map reads to the reference genome (UCSC/mm9). Peak calling and motif analyses were performed using HOMER (v4.9). Heatmap clustering was conducted by deepTools (v3.5.0). Transcription factor footprints were analyzed using Transcription factor Occupancy prediction By Investigation of ATAC-seq Signal (TOBIAS, version 0.15.1) (Bentsen et al, 2020).

## Chromatin immunoprecipitation sequencing (ChIP-seq)

ChIP-seq was performed as described in our previous study (Yokomizo-Nakano et al, 2020). Briefly, cells were fixed by 1.0% paraformaldehyde at 37 °C for 5 min. Fixed cells were lysed and sonicated 15 times at an amplitude of 50% for 10 sec. Samples were incubated with an anti-FLAG antibody (Sigma M2) or anti-HA antibody (Abcam, ab9110) conjugated by Dynabeads protein A/G at 4 °C overnight. ChIP-seq libraries were generated using the ThruPLEX DNA-seq kit (Rubicon Genomics) and then sequenced on NextSeq500 (Illumina). Bowtie2 (v2.2.6) was used to map reads to the reference genome (UCSC/mm9). Peak calling and motif analyses were performed using HOMER (v4.9). Heatmap clustering was conducted by deepTools (v3.5.0).

## RNA sequencing and RamDA sequencing

RNA sequencing was performed as described in our previous study (Yokomizo-Nakano et al, 2020). RamDA sequencing was conducted as previously reported (Hayashi et al, 2018). Briefly, first-strand cDNA was synthesized from a 100-cell lysate using the PrimeScript RT reagent kit (TAKARA Bio) with 1st not-so-random (NSR) primers, and second-strand cDNA was then synthesized using Klenow Fragment (3′-5′ exo-; New England Biolabs) and 2nd NSR primers. Sequencing libraries were generated using the Nextera XT DNA Library Prep kit (Illumina) and sequenced on NextSeq500 (Illumina). Kallisto (v0.43.1) was used to calculate transcripts per million. R software was employed for statistical analyses and a k-means clustering analysis.

## Quantitative RT-PCR (Q-RT-PCR)

Quantitative RT-PCR was performed on LightCycler 480 (Roche) using Luna Universal qPCR Master Mix (New England Biolabs). Expression levels were normalized to those of *B2m*. Primers for PCR were performed by using the following primers: *Hmga2*-Forward 5′-AGGCAGCAAAAACAAGAGCC-3′; *Hmga2*-Reverse 5′-CTGCCTCTTGGCCGTTTTTC-3′; *B2m*-Forward 5′-CTGGCTC ACACTGAATTCACCCC-3′; *B2m*-Reverse 5′-TCGGCCATACTG GCATGCTTAAC-3′; *Igf2bp2*-Forward 5′-TCCCGGGTAGA-CATCCACAGAAA-3′; *Igf2bp2*-Reverse 5′-TCAGCCAGTTTGGT CTCATCAGC-3′; *Hlf*-Forward 5′-GCTTTGCCTTCTGCTCATCT GC-3′; *Hlf*-Reverse 5′-TGCTTTCTCACCTGCCTCCAAC-3′; *Gfi1*-Forward 5′-CAAACACTGATGCCCCCTGA-3′; *Gfi1*-Reverse 5′-ATCCCAAGTCAACCCTGCAA-3′; *Tlr7*-Forward 5′-CTGTTC TACTGGGGTCCAAAGCC-3′; *Tlr7*-Reverse 5′-TGGTTTCCATC-CAGGTAAAGGGC-3′; *Tlr8*-Forward 5′-TGCCAAAGTCTGCTCT CTGCAC-3′; *Tlr8*-Reverse 5′-TGTTTTCCCCTTTCTGGCTGGG-3′; *Ifi204*-Forward 5′-CAGAAGTAACAGGAGAAACATCACT-3′; *Ifi204*-Reverse 5′-GTTGCAGAAGTCTCGCCTCT-3′.

## Western blotting

Whole cell lysates were used for Western blotting. The following antibodies were used: FLAG (Sigma, M2 or Wako, 1E6), Hmga2 (CST, D1A7), HA (Abcam, ab9110), Phos-tag (Fujifilm, BTL-111), and Actin (Santa Cruz Biotechnology, C4).

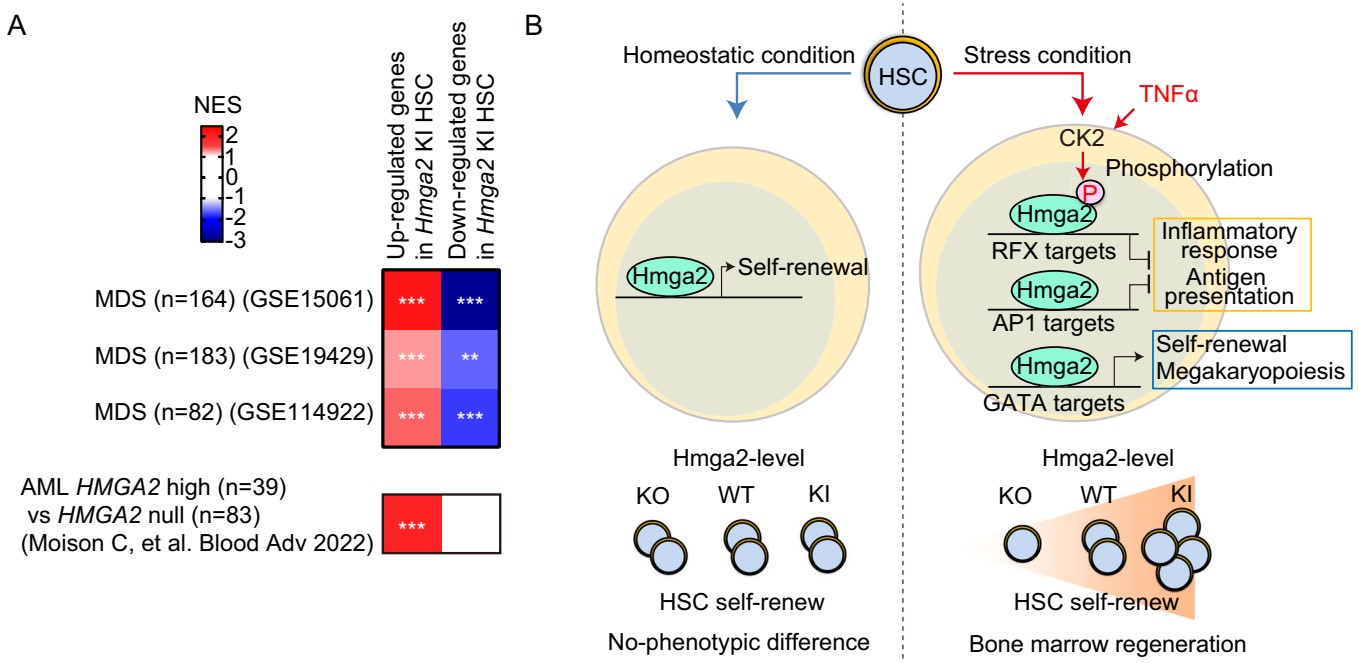

**Figure 7. Stress-induced Hmga2-mediated gene signatures were activated in human MDS HSPCs.**

(A) GSEA of up- and down-regulated genes in *Hmga2* KI HSCs 6 days after the 5-FU injection in CD34[+] HSPCs in patients with MDS relative to those in healthy controls (linked in GSE15061, GSE19429, and GSE114922) and HSPCs in patients with *HMGA2* high AML (RPKM > 2) relative to those with *HMGA2* null AML (RPKM = 0) (linked in GSE49642, GSE52656, GSE62190, GSE66917, GSE67039) (Moison et al, 2022). Asterisks show *FDR < 0.25, **FDR < 0.05, and ***FDR < 0.01. (B) Proposed model for the roles and functions of the Hmga2 protein in normal and stress hematopoiesis.

## Phosphoproteomic analysis

A phosphoproteomic analysis was performed by Proteobiologics (Osaka, Japan). Briefly, $1 \times 10^7$ BM MNCs were pooled from WT mice 3 days after the injection of 250 mg/kg 5-FU. A total of 200ug protein was boiled, reduced, and purified using SP3 protocol (Hughes et al, 2019). Purified protein was digested with trypsin and Lys-C. Desalted peptides were passed through the IMAC/C18 stage-tip, and phosphopeptides were purified on the lower C18 disc. The pooled TMT-labeled phosphopeptides were fractionated by off-line basic pH reversed-phase fractionation using Thermo Scientific UltiMate 3000 UHPLC system. Phosphopeptides were separated by L-column3 C18 column (Chemicals Evaluation and Research Institute), and the LC-MS/MS analysis was conducted by Q Exactive Plus (Thermo Fisher Scientific) coupled with Ultimate 3000 (Thermo Fisher Scientific) and HTC-PAL (CTC Analytics). Phosphopeptide identification and quantification were carried out with MaxQuant 1.6.14.0 supported by the Andromeda search engine, and the cut-off criterion for phosphosite probability was more than 0.75 (Sharma et al, 2014). The statistical analysis was carried out with Perseus 1.6.14.0 (Tyanova et al, 2016).

## Statistical analysis

GraphPad Prism version 9 (GraphPad) was used to perform statistical analyses. The significance of differences was measured by an unpaired two-tailed Student's *t*-test or the Mann–Whitney non-parametric test. A *P* value <0.05 was considered to be significant. No statistical methods were used to predetermine sample sizes for animal studies.

## Data availability

RNA sequencing data were deposited in the DDBJ (https://ddbj.nig.ac.jp/search/en) under the accession numbers: DRA015393, DRA017983, and DRA017984. Sequencing data of DRA013143 were reanalyzed (Sun et al, 2022). ChIP-seq data were deposited in the DDBJ under the accession number: DRA015386. ATAC-seq data were deposited in the DDBJ under the accession number: DRA015385. Detailed information on sequencing data deposited in the DDBJ are shown in Dataset EV9.

The source data of this paper are collected in the following database record: biostudies:S-SCDT-10_1038-S44318-024-00122-4.

## Peer review information

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

## Acknowledgements

This work was supported in part by a grant from the Mochida Memorial Foundation for Medical and Pharmaceutical Research (to SK), the Takeda Science Foundation (to GS and MM), the Princess Takamatsu Cancer Research Fund (to GS), the Daiichi Sankyo Foundation of Life Science (to GS), the Japanese Society of Hematology (to GS), Grant-in-Aid for JSPS Fellows (22KJ2522 to MM), and Grants-in-Aid for Scientific Research (18H02842, 21K19512, 21H02952 (to GS), 21K08421 (to SK), 22K16304 (to JB), and 22H02904 (to MO)) from the Ministry of Education, Culture, Sports, Science and Technology (MEXT) of Japan.

## Author contributions

**Sho Kubota**: Data curation; Formal analysis; Funding acquisition; Validation; Investigation; Methodology; Writing—original draft; Writing—review and editing. **Yuqi Sun**: Investigation. **Mariko Morii**: Investigation; Visualization. **Jie Bai**: Funding acquisition; Investigation. **Takako Ideue**: Resources. **Mayumi Hirayama**: Investigation. **Supannika Sorin**: Investigation. **Eerdunduleng**: Investigation. **Takako Yokomizo-Nakano**: Funding acquisition; Investigation. **Motomi Osato**: Resources; Supervision; Investigation. **Ai Hamashima**: Resources; Supervision; Investigation. **Mihoko Iimori**: Resources; Investigation. **Kimi Araki**: Resources; Supervision; Writing—review and editing. **Terumasa Umemoto**: Conceptualization; Resources; Data curation; Supervision; Funding acquisition; Writing—original draft; Project administration; Writing—review and editing. **Goro Sashida**: Conceptualization; Resources; Data curation; Supervision; Funding acquisition; Investigation; Visualization; Writing—original draft; Project administration; Writing—review and editing.

Source data underlying figure panels in this paper may have individual authorship assigned. Where available, figure panel/source data authorship is listed in the following database record: biostudies:S-SCDT-10_1038-S44318-024-00122-4.

## Disclosure and competing interests statement

The authors declare no competing interests.

# Expanded View Figures

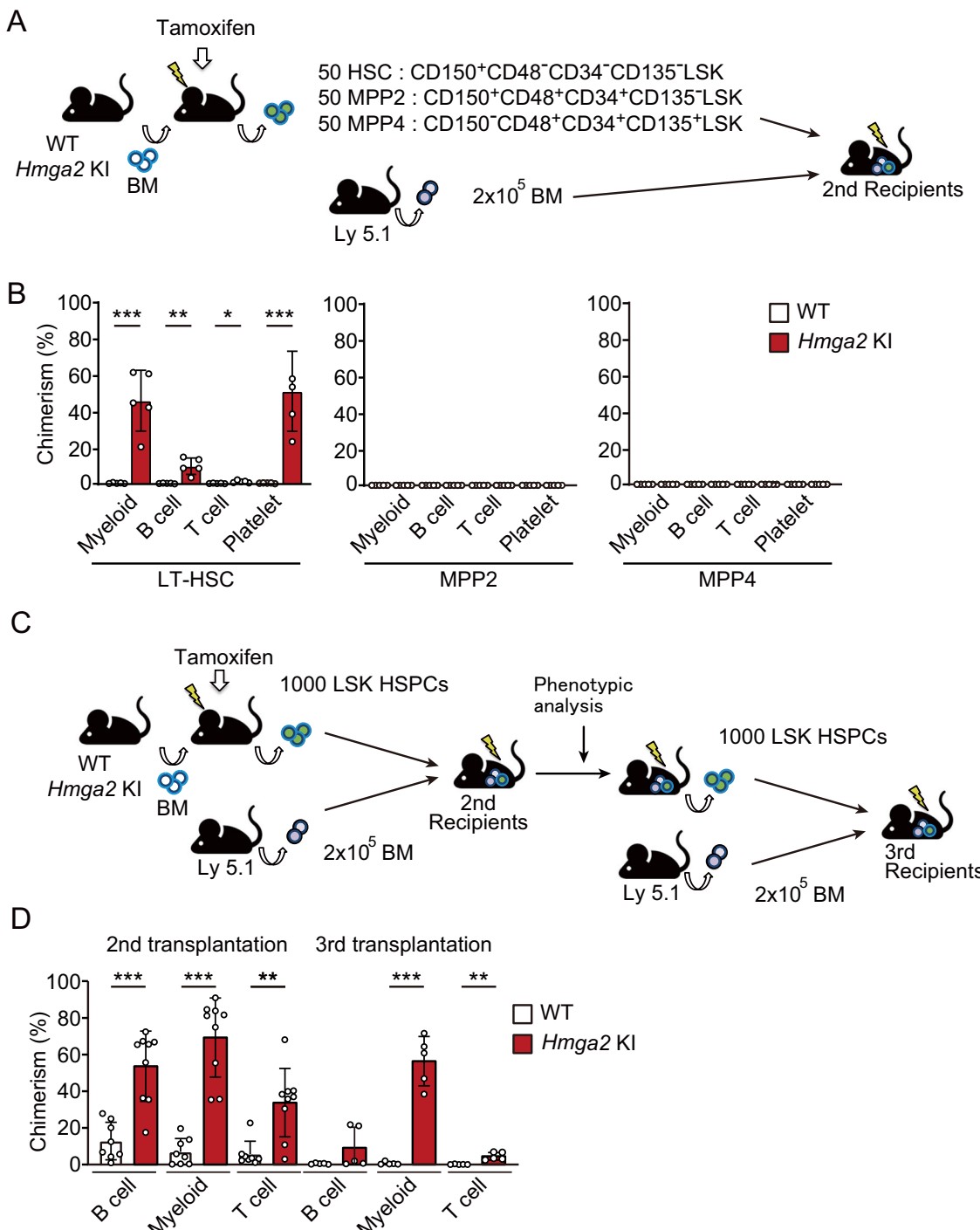

**Figure EV1.  Hmga2 overexpression enhanced the self-renewal of adult HSCs after transplantation.**

(A) Schematic illustration of competitive transplantation showing 50 LT-HSCs, ST-HSCs, or MPP cells (CD45.2+), which were isolated from *Rosa26-YFP* KI or *Hmga2* KI BM cell-transplanted mice 4 months after the tamoxifen treatment, and $2 \times 10^5$ WT BM cells (CD45.1+) were transplanted into lethally-irradiated recipient mice (CD45.1+).
(B) Percentages of CD45.2+ cells in Gr-1+/CD11b+ myeloid cells, B220+ B cells, and CD4+/CD8+ T cells in PB at 4 months (*n* = 5). (C) Schematic illustration of competitive transplantation showing that 1000 LSK HSPCs (CD45.2+), which were isolated from *Rosa26-YFP* KI or *Hmga2* KI BM cell-transplanted mice 4 months after the tamoxifen treatment, and $2 \times 10^5$ WT BM cells (CD45.1+) were transplanted into lethally-irradiated recipient mice (CD45.1+). (D) Percentages of CD45.2+ cells in Gr-1+/CD11b+ myeloid cells, B220+ B cells, and CD4+/CD8+ T cells in PB 4 months after secondary and tertiary transplantations (*n* = 5–9). Data information: In panel (B, D), bars show the mean ± SD, *$p < 0.05$, **$p < 0.01$, and ***$p < 0.001$. *P*-values were calculated by the Student's *t*-test. *N* means the number of mice. Data are representative of two independent experiments.

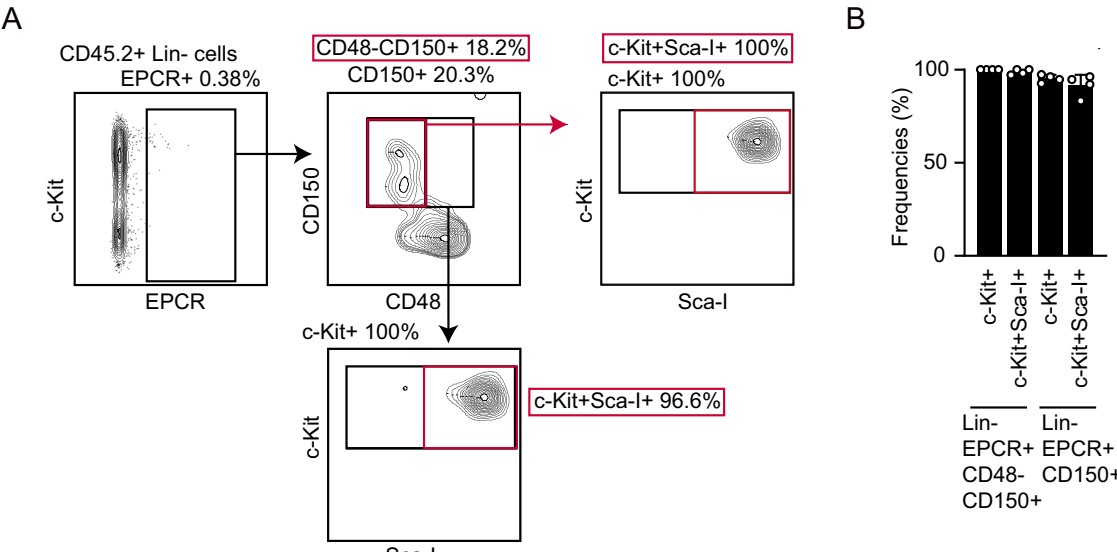

**Figure EV2. Phenotypic definition of HSCs using EPCR.**

(A) Gating strategy of the definition of HSCs and representative flow cytometry plots. (B) Percentages of Kit$^+$ or Kit$^+$Sca-1$^+$ among Lin$^-$EPCR$^+$CD48$^-$CD150$^+$ HSCs and Lin$^-$EPCR$^+$CD150$^+$ cells ($n = 4$; $n$ means the number of mice). Bars show the mean ± SD.

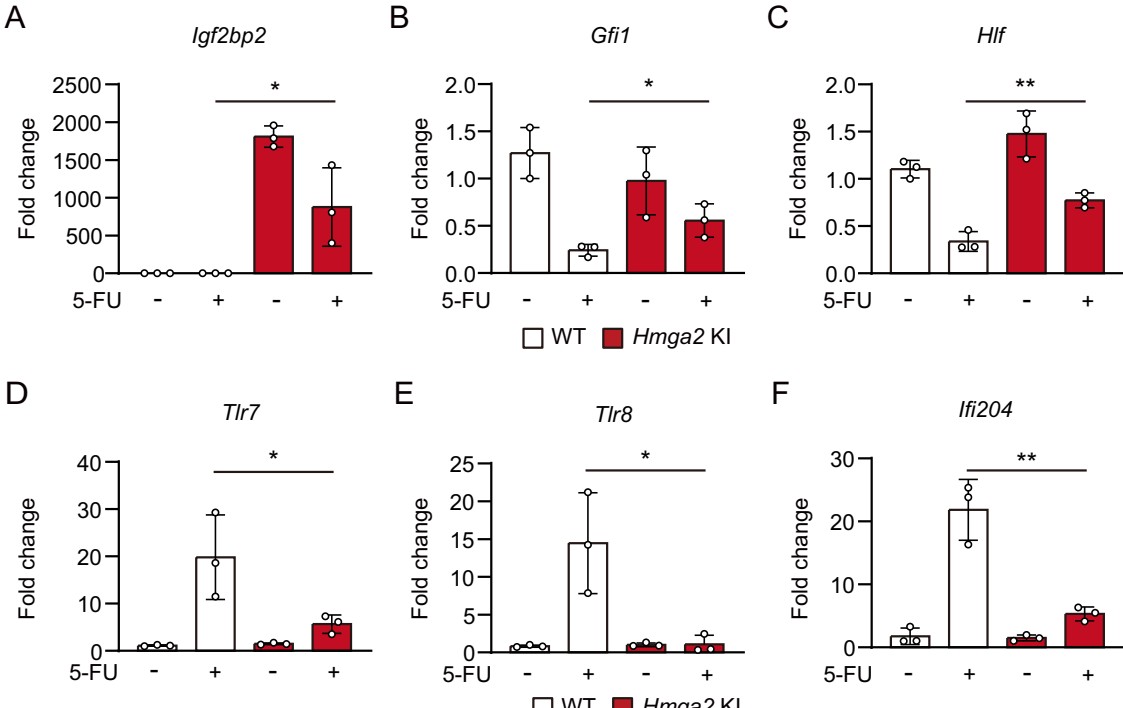

**Figure EV3. Changes in gene expression in HSCs after the 5-FU injection examined by Q-RT-PCR.**

(A–F) Gene expression levels of *Igf2bp2*, *Gfi1*, *Hlf*, *Tlr7*, *Tlr8*, and *Ifi204* in WT and *Hmga2* KI HSCs before and 3 days after the 5-FU injection examined by Q-RT-PCR ($n = 3$). Bars show the mean ± SD, *$p < 0.05$ and **$p < 0.01$. P-values were calculated by the Student's *t*-test. N means the number of samples.

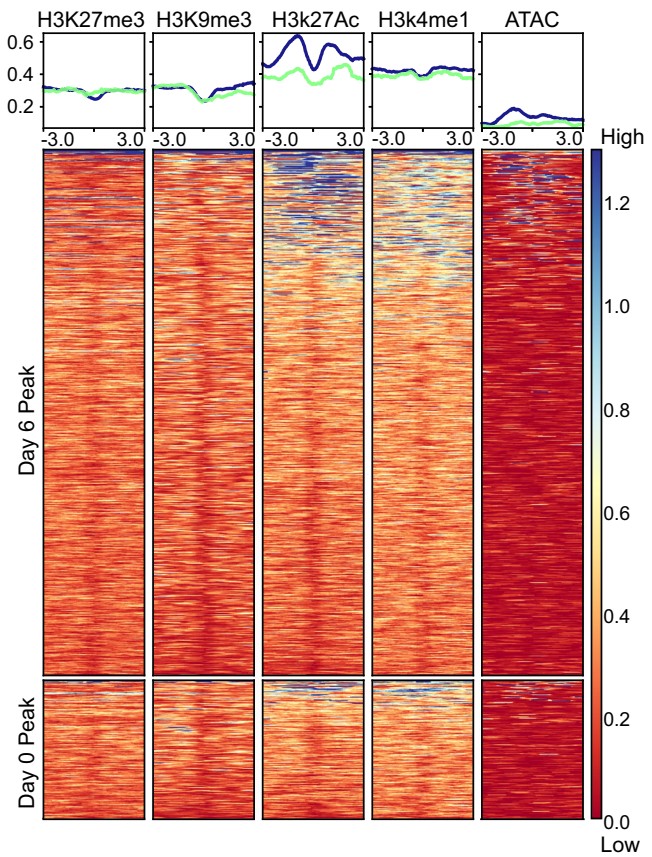

**Figure EV4.   Hmga2 bound to AT-rich and non-regulatory regions in chromatin.**

(**A**) Motif enrichment analyses of Hmga2-ChIP-seq in HSPCs isolated from *HA*-tagged-*Hmga2* KI mice before and 6 days after the injection of 5-FU. Pre-immune IgG was used as a negative control. *P*-values were calculated by using gene ontology functions of HOMER software. (**B**) Heatmaps showing the levels of described histone modifications, such as H3K4me1, H3K9me3, H3K27ac and H3K27me3, and ATAC peaks identified in HSCs (linked in GSE119198, GSE60103, and E-MTAB-11865) in Hmga2-binding regions (Hmga2-binding site ± 3.0 kb) in HSPCs before and after the injection of 5FU.

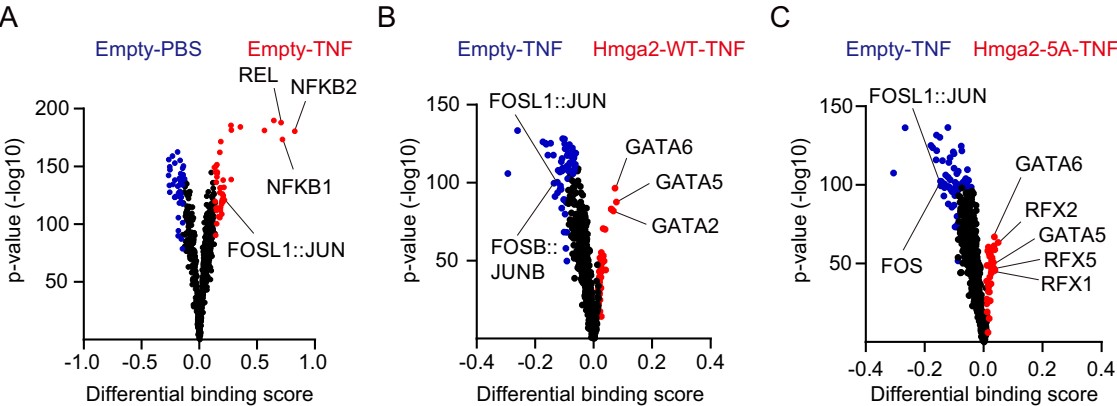

**Figure EV5.  Motif analysis of the open-chromatin region in each comparison.**

(A–C) Volcano plots showing the differential binding activity of TFs between the indicated HSCs. Differentially binding score, transcription factor, and *p*-values were calculated by using TOBIAS software.

