## [Peer Review File · The EMBO Journal]

Chromatin modifier Hmga2 promotes adult hematopoietic stem cell function and blood regeneration in stress conditions

Sho Kubota, Yuqi Sun, Mariko Morii, Jie Bai, Takako Ideue, Mayumi Hirayama, Supannika Sorin, Eerdunduleng, Takako Yokomizo-Nakano, Motomi Osato, Ai Hamashima, Mihoko Imori, Kimi Araki, Terumasa Umemoto, and Goro Sashida

Corresponding author: Goro Sashida (sashidag@kumamoto-u.ac.jp)

Review Timeline:

Submission Date:	14th Jul 23
Editorial Decision:	3rd Sep 23
Revision Received:	14th Mar 24
Editorial Decision:	6th Apr 24
Revision Received:	22nd Apr 24
Accepted:	25th Apr 24

Editor: Daniel Klimmeck

Transaction Report:

Dear Dr Sashida,

Thank you for submitting your manuscript for consideration by the EMBO Journal. Please accept my sincere apologies for getting back to you with unusual protraction due to delayed referee input, as well as detailed discussion in the editorial team. As indicated earlier, your manuscript has been seen by three referees with expertise in HSC and chromatin biology, and we have received reports from all of them, which are shown below.

As you will see from their comments, the referees acknowledge the potential interest and value of your findings for the field. However, they also express major concerns, which need to be addressed thoroughly to make them supportive of publication in the EMBO Journal. In more detail, referee #1 points to related preceding results on the role of HMGA2 in HSC transplantation and therefore states the need for you to expand the current analysis by adding complementary *in vivo* stress data (ref#1, pt.5) and more rigorous analysis of the kinetics upon 5-FU (ref#1, pts. 3,4). This referee is also critical the experimental transplantation set-up chosen (ref#1, pt.1). Further, reviewer #2 requests assessing HSC populations in a different manner to rule out confounding effects of co-purified progenitors (ref#2, pts.1,3). Both referees #2 and #3 find that more work on functional implication of CK2 in this setting is required to make a convincing case (ref#2, pt.4; ref#3, pts. A,B,) and request more MDS signature work (ref#2, pt.5; ref#3.pt. F).

Given the overall interest stated and broader angle of your findings, we are able to invite you to revise your manuscript experimentally to address the referees' comments, along the lines sketched in your outline. I need to stress though that we do require strong support from the referees on a revised version of the study in order to move on to publication of the work. As to the open outcome of the major revisional work required, I suggest keeping EMBO Reports in mind for this study as an alternative venue.

Please feel free to contact me if you have any questions or need further input on the referee comments.

When submitting your revised manuscript, please carefully review the instructions below.

Please feel free to approach me any time should you have additional questions related to this.

Thank you for the opportunity to consider your work for publication.

I look forward to your revision.

Best regards,

Daniel Klimmeck

Daniel Klimmeck, PhD
Senior Editor
The EMBO Journal

Instruction for the preparation of your revised manuscript:

- 1) a .docx formatted version of the manuscript text (including legends for main figures, EV figures and tables). Please make sure that the changes are highlighted to be clearly visible.
- 2) individual production quality figure files as .eps, .tif, .jpg (one file per figure).
- 3) a .docx formatted letter INCLUDING the reviewers' reports and your detailed point-by-point response to their comments. As part of the EMBO Press transparent editorial process, the point-by-point response is part of the Review Process File (RPF), which will be published alongside your paper.
- 4) a complete author checklist, which you can download from our author guidelines (<https://wol-prod-cdn.literatumonline.com/pb->

assets/embo-site/Author Checklist%20-%20EMBO%20J-1561436015657.xlsx). Please insert information in the checklist that is also reflected in the manuscript. The completed author checklist will also be part of the RPF.

6) It is mandatory to include a 'Data Availability' section after the Materials and Methods. Before submitting your revision, primary datasets produced in this study need to be deposited in an appropriate public database, and the accession numbers and database listed under 'Data Availability'. Please remember to provide a reviewer password if the datasets are not yet public (see <https://www.embopress.org/page/journal/14602075/authorguide#datadeposition>).

7) Our journal encourages inclusion of *data citations in the reference list* to directly cite datasets that were re-used and obtained from public databases. Data citations in the article text are distinct from normal bibliographical citations and should directly link to the database records from which the data can be accessed. In the main text, data citations are formatted as follows: "Data ref: Smith et al, 2001" or "Data ref: NCBI Sequence Read Archive PRJNA342805, 2017". In the Reference list, data citations must be labeled with "[DATASET]". A data reference must provide the database name, accession number/identifiers and a resolvable link to the landing page from which the data can be accessed at the end of the reference. Further instructions are available at .

8) At EMBO Press we ask authors to provide source data for the main and EV figures. Our source data coordinator will contact you to discuss which figure panels we would need source data for and will also provide you with helpful tips on how to upload and organize the files.

Numerical data can be provided as individual .xls or .csv files (including a tab describing the data). For 'blots' or microscopy, uncropped images should be submitted (using a zip archive or a single pdf per main figure if multiple images need to be supplied for one panel). Additional information on source data and instruction on how to label the files are available at .

9) We replaced Supplementary Information with Expanded View (EV) Figures and Tables that are collapsible/expandable online (see examples in <https://www.embopress.org/doi/10.15252/emj.201695874>). A maximum of 5 EV Figures can be typeset. EV Figures should be cited as 'Figure EV1, Figure EV2' etc. in the text and their respective legends should be included in the main text after the legends of regular figures.

11) For data quantification: please specify the name of the statistical test used to generate error bars and P values, the number (n) of independent experiments (specify technical or biological replicates) underlying each data point and the test used to calculate p-values in each figure legend. The figure legends should contain a basic description of n, P and the test applied. Graphs must include a description of the bars and the error bars (s.d., s.e.m.).

The revision must be submitted online within 90 days; please click on the link below to submit the revision online before 2nd Dec 2023.

Referee #1:

Even though the chromatin modifier Hmga2 is highly expressed in HSCs, under homeostasis a lack or overexpression of Hmga2 does not lead to a stem cell phenotype. Here the authors used 5FU treatments to induce stress in KO or KI mice and report enhanced self renewal in KI mice, and decrease in HSC numbers and delayed recovery in KO mice, thus indicating a role for Hmga2 in the stress response of HSCs. Mechanistically they link this to the TNFa-CK2-Hmga2 signaling axis.

Major points:

1. The authors have already reported in previous publications that Hmga2 overexpression results in increased self-renewal of HSCs in a transplant, and thus stress setting. Here they use 5-FU, another stress inducer. However, the observation that Hmga2 plays a role in the stress response of HSCs is thus not novel
2. A main concern is also that the mouse experiments are all performed on a transplantation background, which, as the authors also state in their intro and discussion, is creating a stress situation for HSCs as well, including an increased inflammatory environment. The timing in the experiments is also not clear; what was the time between transplant and tamoxifen, between tamoxifen and 5-FU or between transplant and 5-FU?
3. WT mice do recover from 5-FU as well, however it looks from the data in figure 1 that there is especially a difference in the recovery time. Thus comparing signatures and profiles at the same time point after 5-FU treatment might miss this point in differences in the recovery time. It would have been good to really look into the dynamics of the response and compare the dynamic changes between KI and WT, because changes in KI might simply come up later in WT HSCs because the delay in response. The same is true for the comparison WT and KO, where it looks like the KO show a delayed recovery compared to the WT (Figure 2).
4. For several figure panels the number of data points is very limited: only 2 or 3. For the RNAseq, ATACseq and CHIPseq it is not always clear how many biological replicates were used, and if it is stated it is only 2.
5. From Figure 4 on they change their stress induced treatment from 5-FU to TNFa, however only in in vitro settings, as far as I can see. Have the authors also performed in vivo TNFa treatments on the different mice? This would also strengthen the story, not only showing one type of stress. 5-FU is not a natural form of stress to the system. Inflammation for example would be. This would also strengthen the potential link the authors would want to make with diseased states such as MDS.
6. Many statements are very strong, whereas for the majority of the data they show a correlation, not a functional proof. The link with TNFa for example could be strengthened by experiments in which TNFa signaling would be blocked, either using KO or inhibitors. Based on the current data statements should be softened, speaking more about 'indicating, suggesting' etc, rather than 'show, proof'

Minor points:

- The text and abstract would really benefit from an English correction. Some of the words used are not very accurate and several sentences do not really flow.
- Suppl. Fig 2: a) the figures says 50 cells, the figure legend 200? What was the background of the 2nd recipients? And in the 2nd and 3rd round of transplants one would also still expect WT HSCs to show chimerism, why is this not the case (Suppl. Fig 2D)?
- Figure 1G: is this difference in cell cycle significant? Based on the error bars I have some doubts.
- Figure 1H and 1I: I don't understand how this assay works and what it will show you. Could you explain this one better, please?
- Figure 2: is there also a difference in cell cycle/proliferation between WT and KO HSCs?
- Figure 3: the RNAseq is performed only 2 months post transplantation, when the system is not fully recovered yet from the transplant. It is also not clear why the authors have chosen 3 and 6 days post 5FU as timepoints of analysis. A validation of the RNAseq data, for example by qPCR of the key-changed genes, would have been good.
- Figure 4E-F: it is known that TNFa directly impacts on WT HSCs. Why is there a change upon TNFa treatment in wt HSCs in 4F but not in 4E?
- Figure 4I: also WT HSCs do proliferate upon 5-FU (Fig 1G), so why are KI enriched for cell cycle and proliferation pathways? Could this be a timing in the stress response dynamics? Which genes are behind the GO terms of cluster 1?
- Figure 4J: It would be interesting to include the WT data in here as well, since they should be the intermediate between KO and KI. In fact, if Hmga2 is a key factor, would you even expect to see overlap between KO and KI? Or does this speak again for a

delay in response dynamics?

- Suppl Figure 1: why is this not labeled suppl Fig 4 in between the current Suppl Fig 3 and 4 to stick to the order of when the different figures are discussed in the text?
- Figure 5: b) the Phostag input is overexposed, impossible to say something. C) it looks like the lower band in the anti-Flag is running lower in the Hmga2-5A sample?
- Figure 5G: why are WT HSCs used here? Don't you introduce an overexpression of Hmga2 then?
- In the text on page 16, line 7 there is a reference to Figure 5F, is this correct?
- Figure 5N: is this really significant with such a data spread?
- Figure 6: the link to MDS is interesting but the conclusion only based on a signature comparison highly overstated. The stress-induced Hmga2 signature is based on 5-FU treatment, that is not a natural form of stress or inflammation. One should keep that in mind as well!

Referee #2:

Kubota and colleagues examined the role of High mobility group AT-hook 2 (Hmga2) in adult stress hematopoiesis. The author claimed that Hmga2 was crucial to hematopoietic stem and progenitor cells in response to stress during hematopoietic reconstruction. Hmga2 was shown to activate stem-cell maintenance genes while suppressing inflammatory genes due to chromatin remodeling. Interestingly, the observed chromatin remodeling depended on the proper phosphorylation of Hmga2 at its C-terminus. Abruption of Hmga2 phosphorylation diminished HSC expansion under inflammatory stress. The author further speculated on the potential role of Hmga2 in MDS development. This manuscript attempted to address the unidentified function of Hmga2 in adult hematopoiesis. While this manuscript implied an interesting and significant role of Hmga2 in adult stress hematopoiesis and disease development, more evidence needs to be provided to corroborate the findings. Thus, major revisions are recommended before considering publication.

Major concerns:

1. In the models involving 5FU challenges, HSCs were defined as Lin-EPCR+CD48-CD150+. While it is true that 5FU treatment reduces the cKIT expression on HSPCs, it is almost certain that this population contains HSCs and other progenitor cells, especially during the later stage of hematopoietic regeneration. While it is appreciated that the author tried to define phenotypical HSCs consistently, the populations examined at different stages are different (prior injection vs Day 3 and Day 6). Thus, it is unknown if the increased number of "HSCs" was truly due to HSC itself (Figure 1D). This speculation can be indirectly corroborated by Figure 3B. It is recommended that Lin-cKIT+EPCR+CD48-CD150+ HSCs should be examined for their number, frequency, and cell cycle status at least 12 days post-5FU to strengthen the statement about HSCs.
2. The author claimed that Hmga2 KI HSCs were undergone symmetric division based on the observation where 50% more HSCs were identified after the first division (Figure 1H-I). This statement could only be valid if Hmga2 KI HSCs and WT HSCs were under the same cell cycle kinetics, the same vulnerability of cell death, and the same rate of HSC homing (because this experiment was based on transplantation). This premise, however did not seem entirely true based on Figure 1G (where the cell cycle was different between WT and Hmga2 KI HSCs). And the effect of homing and cell death was not examined. Thus the increased number of labeled Hmga2 KI HSCs could result from many factors. Cell cycle kinetics, homing, and cell death should be examined. Additionally, imaging should be performed to provide direct evidence of symmetric vs asymmetric division of HSCs.
3. It is possible that Hmga2 expression is distinct across HSPC populations. Even if its expression was comparable, the chromatin binding site of Hmga2 could be highly cell-type dependent. As the possibility of varied HSPC populations between control and Hmga2 KI at different timepoints of 5FU challenge, it is not certain if the changes observed in Figure 4G-H truly reflect the changes in Hmga2 binding, or it is merely the consequence of different HSPC populations were sampled. It is recommended to quantify Hmga2 expression across various HSPC populations and its binding in the purified HSC population. The same concern also stands for Figure 5F-H.
4. Does Casein kinase 2 express in HSCs? Could it be possible that CK2 levels are changed in response to 5FU/stress?
5. It is unclear whether Hmga2 overexpression was evident in MDS, although GSEA analysis was performed (Figure 6A). Hmga2 levels should be examined in the age-matched MDS and healthy control HSPCs. This could be easily done using publicly available microarray data.
6. It will also be helpful to examine Hmga2 expression over aging in WT mice.

Minor concerns:

1. Inconsistent information provided in different sections made it difficult to evaluate some experiments. For instance, tamoxifen

treatment to induce Hmga2 expression in the KI model was described as one month in Result Section 1 (Page 9, line 7-8). However, the same experiment was depicted as 5 consecutive days in the Methods Mice (Page 5, line 6).

2. While no information was provided, it was assumed that no hematopoietic contribution could be observed 12 days post-FU for figures in Figure 1? It will be clear if chimerism can be quantified.
3. Could it be possible to provide targeting (KO) efficiency of Hmga2 KO mouse?
4. Are control mice used for Figure 1 and Figure 2 the same? Parameters, including platelet numbers and HSC frequencies, are dramatically different. Could it be possible to provide potential reasons?
5. Given that Hmga2 may suppress inflammatory genes, could this be examined by measuring plasma levels of the same cytokines (Figure 4C) in 5FU-treated Hmga2a-KI mice?

Referee #3:

The authors report an important role for Hmga2 downstream of CK2 and TNF α in stress hematopoiesis. The story is of interest and the authors need to clarify and better connect a few aspects of the story together as it is outlined below:

- A) What are the CK2 levels before and after 5-FU treatment in HSCs? What are the CK2 substrates before and after treatment of HSCs with 5-FU? Is Hmga2 one of the top substrates? Are other HMG members substrates CK2 substrates as well? Phosphoproteomics analysis should be performed here.
- B) Can the phenotype observed upon 5-FU (in the case of Hmga2 WT or KI) be rescued by CK2 silencing or the use of CK2 inhibitors?
- C) How is Hmga2 protein expression regulated (increased) upon 5-FU treatment? Is it through decreased protein turnover via deubiquitination and block of proteasomal degradation?
- D) "We next performed RNA sequencing of WT and Hmga2 cKO Lineage-CD150+CD48-EPCR+ HSCs on days 0 and 6 after the 5-FU injection. We identified 1151 up-regulated genes and 717 down-regulated genes in Hmga2 cKO HSCs, relative to those in WT HSCs 6 days after the injection." How does the total amount of RNA isolated from the same number of cells compare between the different sets? I was wondering whether the authors need to use spike-in RNA to normalize their analysis in case the amount of RNA differs a lot.
- E) Can ectopic expression of RFX5 rescue the Hmga2-mediated phenotype?
- F) The connection of the findings with MDS is rather loose. The authors need to discuss more about specific genes, including RFX5.

The paper by Moison et al., Blood Adv. 2022 Aug 23 (AML setting) might be an interesting study and the authors could compare their gene expression signatures to the signatures in this paper.

Referee #1:

Even though the chromatin modifier *Hmga2* is highly expressed in HSCs, under homeostasis a lack or overexpression of *Hmga2* does not lead to a stem cell phenotype. Here the authors used 5FU treatments to induce stress in KO or KI mice and report enhanced self renewal in KI mice, and decrease in HSC numbers and delayed recovery in KO mice, thus indicating a role for *Hmga2* in the stress response of HSCs. Mechanistically they link this to the TNF α -CK2-*Hmga2* signaling axis.

Thank you for your time and comments, which have significantly improved the quality of the revised manuscript. We performed the additional experiments requested and added the data obtained to the revised manuscript.

Major points:

1. The authors have already reported in previous publications that *Hmga2* overexpression results in increased self-renewal of HSCs in a transplant, and thus stress setting. Here they use 5-FU, another stress inducer. However, the observation that *Hmga2* plays a role in the stress response of HSCs is thus not novel

Response: Consistent with previous studies showing *Hmga2*-induced self-renewal in tissue stem cells (Nishino J, et al. Cell 2008; Copley MR, et al. Nature Cell Biology 2013), we demonstrated that the overexpression of *Hmga2* increased the self-renewal of adult hematopoietic stem cells (HSCs) after transplantation, but did not induce hematological malignancies in mice (Sun, et al. Int J Hematol, 2022). We selected 5-FU to examine the behavior of HSCs in mice in the early phase during bone marrow (BM) regeneration when HSCs expand and generate progenitor cells. The molecular mechanisms by which *Hmga2* increases the self-renewal capacity of HSCs through the TNF α -CK2-*Hmga2* axis have not yet been demonstrated elsewhere.

2. A main concern is also that the mouse experiments are all performed on a transplantation background, which, as the authors also state in their intro and discussion, is creating a stress situation for HSCs as well, including an increased inflammatory environment. The timing in the experiments is also not clear; what was the time between transplant and tamoxifen, between tamoxifen and 5-FU or between transplant and 5-FU?

Response: Since the Reviewer expressed a concern regarding the inflammatory condition induced by the tamoxifen injection and/or transplantation, we added this as a limitation to page 19 of the revised manuscript. In addition, we apologize for the lack of clear information on the timing of these treatments. We added more detailed descriptions of the timing of the tamoxifen injection and transplantation to the figure legends and illustrations for those experiments to Figures 1A, 2C, 4G, 5D, 6B, and 6K in the revised manuscript. Briefly, all experiments were performed using mice one month after the injection of tamoxifen to induce the expression of *Hmga2* or the deletion of the *Hmga2* gene. In 5-FU experiments, these bone marrow cells were transplanted into lethally irradiated mice, and we injected 5-FU into recipient mice 6 weeks after transplantation.

3. WT mice do recover from 5-FU as well, however it looks from the data in figure 1 that there is especially a difference in the recovery time. Thus comparing signatures and profiles at the same time point after 5-FU treatment might miss this point in differences in the recovery time. It would have been good to really look into the dynamics of the response and compare the dynamic changes between KI and WT, because changes in KI might simply come up later in WT HSCs because the delay in response. The same is true for the comparison WT and KO, where it looks like the KO show a delayed recovery compared to the WT (Figure 2).

Response: As suggested by the Reviewer, we analyzed the dynamics of bone marrow cells after the 5-FU injection. *Hmga2* KI mice showed similar BM cell counts to WT mice before the injection; however, *Hmga2* KI mice showed more rapid increases in BM cell counts and HSC frequencies on days 7 and 12 after the 5-FU injection than those in WT mice. Thirty days after the injection, we did not observe any changes in the number of HSCs or BM cells in Figure 1C and 1D in the revised

manuscript (please refer to the panels below), suggesting that the overexpression of *Hmga2* drove the rapid expansion of HSCs and BM cells in the early stage in response to stress; however, after the resolution of stress, the overexpression of *Hmga2* did not appear to affect hematopoiesis in the steady state.

We also analyzed *Hmga2* KO mice after the 5-FU injection and found that BM cell and HSC numbers were lower in *Hmga2* KO mice 12 days after the injection and also that 6 out of 10 *Hmga2* KO mice died within 30 days of the 5-FU injection (please refer to survival data below). This observation suggests that the presence of *Hmga2* facilitated the expansion of HSCs and the recovery of hematopoiesis and also secured the survival of mice after the 5-FU injection, at least in this experimental setting. We added these data to Figure 2F, Figure 2G, and Figure 2I and to page 7-8 in the revised manuscript.

4. For several figure panels the number of data points is very limited: only 2 or 3. For the RNAseq, ATACseq and CHIPseq it is not always clear how many biological replicates were used, and if it is stated it is only 2.

Response: As suggested by the Reviewer, we performed additional experiments or increased the number of replicates of ChIP-seq and RNA-seq data in Figure 4J, 4M-O, 6G, and 6H. We also added samples of CBC, flow cytometry, and cytokine arrays to Figure 1B, 1C, 1D, 1E, 1F, 2E, 4C, and 4F in the revised manuscript. We confirmed the results of RNA-seq in Figure 3B-D by performing Q-RT-PCR on six representative genes, including stem cell genes and inflammatory regulator genes, which are shown in Figure EV3 (these data are described below in response to minor comment #6). We updated information on the numbers of experiments and samples in the corresponding figure legends in the revised manuscript.

5. From Figure 4 on they change their stress induced treatment from 5-FU to TNF α , however only in *in vitro* settings, as far as I can see. Have the authors also performed *in vivo* TNF α treatments on the different mice? This would also strengthen the story, not only showing one type of stress. 5-FU is not a natural form of stress to the system. Inflammation for example would be. This would also strengthen the potential link the authors would want to make with diseased states such as MDS.

Response: As suggested by the Reviewer, we intravenously injected 2 μ g of TNF α three times into

wild-type control mice, *Hmga2* KI mice, and *Hmga2* KO mice, and performed a phenotypic analysis of BM cells two days later. Consistent with the results obtained after the TNF α treatment under *in vitro* conditions, we found that *Hmga2* KI mice showed higher HSC counts in BM than WT mice two days after the injection, while *Hmga2* KO mice showed lower HSC counts in BM, as shown below. We added the results obtained under *in vivo* conditions to Figure 4H and 4I in the revised manuscript (in page 10-11; please refer to the panels below).

6. Many statements are very strong, whereas for the majority of the data they show a correlation, not a functional proof. The link with TNF α for example could be strengthened by experiments in which TNF α signaling would be blocked, either using KO or inhibitors. Based on the current data statements should be softened, speaking more about 'indicating, suggesting' etc, rather than 'show, proof'

Response: As suggested, we changed and modified the sentences accordingly in the revised manuscript.

Minor points:

1- The text and abstract would really benefit from an English correction. Some of the words used are not very accurate and several sentences do not really flow.

Response: The revised manuscript was proofread by a native English speaker.

2- Suppl. Fig 2: a) the figures says 50 cells, the figure legend 200? What was the background of the 2nd recipients? And in the 2nd and 3rd round of transplants one would also still expect WT HSCs to show chimerism, why is this not the case (Suppl. Fig 2D)?

Response: We apologize for our mistake in the original manuscript. Since 50 cells was correct, we corrected the figure legend in the revised manuscript. Recipients were Ly5.1+ C57BL/6-background wild-type mice. In Supplementary Figure 2D in the original manuscript, we transplanted 1000 purified LSK HSPCs, which may provide a greater repopulating capacity, than 50 HSCs alone, which was transplanted into recipients in Supplementary Figure EV1 in the revised manuscript.

3- Figure 1G: is this difference in cell cycle significant? Based on the error bars I have some doubts.

Response: To clearly show these differences, we provide individual panels of each cell cycle phase, which formed the bar graph in Figure 1G in the revised manuscript.

4- Figure 1H and 1I: I don't understand how this assay works and what it will show you. Could you explain this one better, please?

Response: We attempted to improve its explanation on page 7 in the revised manuscript. Briefly, CytoTell labeling is a dye for a lipophilic membrane similar to CFSE that is often used to measure the number of cell divisions by dye dilution. We previously showed that HSC proliferation started 4 days after the 5-FU injection at the same dosage as that used in the present study (Umemoto T, et al. EMBO J, 2022). When we transplanted CytoTell-labeled HSCs into mice 4 days after the injection, we were able to measure how many HSCs remained after the first division *in vivo* under stress conditions.

5- Figure 2: is there also a difference in cell cycle/proliferation between WT and KO HSCs?

Response: As requested, we performed a cell cycle analysis of WT and *Hmga2* KO HSCs 12 days after the 5-FU injection, and found that a slightly higher percentage of *Hmga2* KO HSCs were in the G1 stage than WT HSCs. Please refer to the panel below.

6- Figure 3: the RNAseq is performed only 2 months post transplantation, when the system is not fully recovered yet from the transplant. It is also not clear why the authors have chosen 3 and 6 days post 5FU as timepoints of analysis. A validation of the RNAseq data, for example by qPCR of the key-changed genes, would have been good.

Response: As indicated by the Reviewer, hematopoiesis had not fully recovered 2 months after transplantation; however, we did not detect marked changes in the transcriptome in HSCs or MPP between WT and *Hmga2* KI mice in Figure 3A.

In response to the second comment, we previously examined the recovery of HSCs after a 5-FU injection, and demonstrated that HSCs started to divide from 3 to 4 days and continuously divided while maintaining stem cell features until 6 days after the injection (Umemoto et al, J Exp Med, 2018; Umemoto T, et al. EMBO J, 2022). These findings suggest that days 3 or 6 after the 5-

FU injection, which was used at the same dosage as in these studies, are critical time points when HSCs start or promote self-renewing divisions, respectively. Therefore, we selected 3 and 6 days after the injection in the present manuscript. To support this choice for studying the dynamics of HSCs under stress, we performed Q-RT-PCR on the genes labeled in Figure 3C, and found that *Hmga2* KI HSCs activated stem cell genes, but repressed genes involved in the inflammatory pathway, such as *Tlr7*, *Tlr8*, and *Ifi204*, 3 days after the injection. We added these Q-RT-PCR data to Supplementary Figure EV3 in the revised manuscript, as also shown below.

7- Figure 4E-F: it is known that TNF α directly impacts on WT HSCs. Why is there a change upon TNF α treatment in wt HSCs in 4F but not in 4E?

Response: We understand the Reviewer's concern regarding WT cells. We repeated the experiments in Figure 4F, and found that the number of WT HSCs slightly decreased after the TNF- α treatment, while *Hmga2* KO HSCs markedly decreased. In Figure 4E, the number of WT HSCs slightly decreased after the TNF- α treatment, suggesting that under this condition, TNF- α slightly repressed WT HSCs despite a difference in the degree of the reduction in WT HSCs in these figures. This difference may be partly attributed to the different number of cells at the start of the culture; 500 cells in Figure 4E and 1000 cells in Figure 4F were sorted in a well. We added this information to the figure legends on page 34 in the revised manuscript:

8- Figure 4I: also WT HSCs do proliferate upon 5-FU (Fig 1G), so why are KI enriched for cell cycle and proliferation pathways? Could this be a timing in the stress response dynamics? Which genes are behind the GO terms of cluster 1?

Response: As indicated by the Reviewer, *Hmga2* KI HSCs enriched the expression of genes involved in proliferation pathways 6 days after the injection, forming cluster 2. We observed more cycling cells in *Hmga2* KI HSCs than in WT HSCs 7 days after the injection in Figure 1G. This difference in the enrichment of genes may be caused by the timing of the response to the stress and the different expression levels of genes, which may account for the difference observed in the behavior of HSCs between *Hmga2* KI and WT mice. Regarding cluster 1 genes, the expression of genes involved in cell adhesion and infection pathways was higher in WT HSCs than in *Hmga2* KI HSCs, which may reflect the activation of inflammation pathways identified by GSEA using the transcriptome of all genes in Figure 3F. We listed GO terms in all clusters in Figure 4K in Dataset-EV4 in the revised manuscript.

9- Figure 4J: It would be interesting to include the WT data in here as well, since they should be the intermediate between KO and KI. In fact, if *Hmga2* is a key factor, would you even expect to see overlap between KO and KI? Or does this speak again for a delay in response dynamics?

Response: We want to explain Figure 4M in the revised manuscript; the panel showed comparisons between down-regulated genes in *Hmga2* KI HSCs and up-regulated genes in *Hmga2* KO relative to those in WT HSCs six days after the injection, suggesting that *Hmga2* suppressed the expression of genes involved in inflammatory pathways after the stress.

As suggested by the Reviewer, we drew a Venn diagram showing up-regulated genes in *Hmga2* KI HSCs, *Hmga2* KO HSCs, and WT HSCs six days after the injection relative to the corresponding genotype HSCs before the injection, and another Venn diagram showing down-regulated genes in *Hmga2* KI HSCs, *Hmga2* KO HSCs, and WT HSCs six days after the injection relative to the corresponding genotype HSCs before the injection. As expected, WT HSCs showed an intermediate transcriptional profile between KO and KI after the injection (please refer to the panel below).

10- Suppl Figure 1: why is this not labeled suppl Fig 4 in between the current Suppl Fig 3 and 4 to stick to the order of when the different figures are discussed in the text?

Response: We corrected this in the revised manuscript.

11- Figure 5: b) the Phostag input is overexposed, impossible to say something. C) it looks like the lower band in the anti-Flag is running lower in the *Hmga2*-5A sample?

Response: We provided short-expose pictures in Figure 5B and Appendix Figure S4 in the revised manuscript. There was no change in the protein sizes of *Hmga2*-WT and *Hmga2*-5A without the CK2 reaction. In contrast, *Hmga2*-WT shifted after the *in vitro* kinase reaction with CK2, whereas the *Hmga2*-5A mutant did not (please refer to the panels below), suggesting that the high levels of phosphorylation induced by CK2 shifted the band of *Hmga2*-WT.

12- Figure 5G: why are WT HSCs used here? Don't you introduce an overexpression of *Hmga2* then?

Response: We understand the Reviewer's concern and think that it may be better to use *Hmga2* KO cells. Since we employed virus-vector transduction, the expression levels of exogenous wild-type or mutant *Hmga2* mRNA were approximately 100-fold higher than those of endogenous *Hmga2*. Therefore, the contribution of endogenous *Hmga2* was likely to be negligible in these transduced cells.

13- In the text on page 16, line 7 there is a reference to Figure 5F, is this correct?

Response: This is correct. It was moved to Figure 6C on page 13 in the revised manuscript.

14- Figure 5N: is this really significant with such a data spread?

Response: This is significant because the P-value of empty-control-WT versus empty-control-*Hmga2* KO was $p=0.0175$; empty-control *Hmga2* KO versus *Hmga2*-WT-transduced *Hmga2* KO was $p=0.0067$; *Hmga2*-WT-transduced *Hmga2* KO versus *Hmga2*-5A-transduced *Hmga2* KO was $p=0.0134$, which are shown in Figure 6J in the revised manuscript.

15- Figure 6: the link to MDS is interesting but the conclusion only based on a signature comparison highly overstated. The stress-induced *Hmga2* signature is based on 5-FU treatment, that is not a natural form of stress or inflammation. One should keep that in mind as well!

Response: We agree and modified the sentences on page 15-16 in the revised manuscript accordingly.

Referee #2:

Kubota and colleagues examined the role of High mobility group AT-hook 2 (*Hmga2*) in adult stress hematopoiesis. The author claimed that *Hmga2* was crucial to hematopoietic stem and progenitor cells in response to stress during hematopoietic reconstruction. *Hmga2* was shown to activate stem-cell maintenance genes while suppressing inflammatory genes due to chromatin remodeling. Interestingly, the observed chromatin remodeling depended on the proper phosphorylation of *Hmga2* at its C-terminus. Abruption of *Hmga2* phosphorylation diminished HSC expansion under inflammatory stress. The author further speculated on the potential role of *Hmga2* in MDS development. This manuscript attempted to address the unidentified function of *Hmga2* in adult hematopoiesis. While this manuscript implied an interesting and significant role of *Hmga2* in adult stress hematopoiesis and disease development, more evidence needs to be provided to corroborate the findings. Thus, major revisions are recommended before considering publication.

We thank you for your time and comments. We performed the additional experiments requested and added the data obtained to the revised manuscript.

Major concerns:

1. In the models involving 5FU challenges, HSCs were defined as Lin-EPCR+CD48-CD150+. While it is true that 5FU treatment reduces the cKIT expression on HSPCs, it is almost certain that this population contains HSCs and other progenitor cells, especially during the later stage of hematopoietic regeneration. While it is appreciated that the author tried to define phenotypical HSCs consistently, the populations examined at different stages are different (prior injection vs Day 3 and Day 6). Thus, it is unknown if the increased number of "HSCs" was truly due to HSC itself (Figure 1D). This speculation can be indirectly corroborated by Figure 3B. It is recommended that Lin-cKIT+EPCR+CD48-CD150+ HSCs should be examined for their number, frequency, and cell cycle status at least 12 days post-5FU to strengthen the statement about HSCs.

Response: This is an important question. The Reviewer pointed out that c-Kit expression was

reduced after the 5-FU treatment, and we consistently demonstrated that c-Kit is not a definitive marker for HSCs in the early stage of BM regeneration (Umemoto T, et al. J Exp Med 2018; Umemoto T, et al. EMBO J 2022). Our previous studies showed that the engraftment activity of HSCs with reduced c-Kit levels after the administration of 5-FU was similar to that of HSCs showing high c-Kit expression levels in the steady state. This finding suggests that the down-regulation of c-Kit during BM regeneration does not always reflect the loss of stemness. Moreover, the down-regulation of c-Kit has been attributed to BM environmental changes after the 5-FU injection. Importantly, even in the steady state, Lin-EPCR+CD48-CD150+ cells were mostly positive for c-Kit (Umemoto T, et al. J Exp Med 2018; Umemoto T, et al. EMBO J 2022), as shown below, supporting Lin-EPCR+CD48-CD150+ being HSCs. Therefore, we selected the definition of HSCs without relying on c-Kit (Lin-EPCR+CD48-CD150+ cells) to precisely identify *bona fide* HSCs, at least during BM regeneration after the 5-FU injection. We added this information to Supplementary Figure EV2 in the revised manuscript (please refer to the panels below).

2. The author claimed that *Hmga2* KI HSCs were undergone symmetric division based on the observation where 50% more HSCs were identified after the first division (Figure 1H-I). This statement could only be valid if *Hmga2* KI HSCs and WT HSCs were under the same cell cycle kinetics, the same vulnerability of cell death, and the same rate of HSC homing (because this experiment was based on transplantation). This premise, however did not seem entirely true based on Figure 1G (where the cell cycle was different between WT and *Hmga2* KI HSCs). And the effect of homing and cell death was not examined. Thus the increased number of labeled *Hmga2* KI HSCs could result from many factors. Cell cycle kinetics, homing, and cell death should be examined. Additionally, imaging should be performed to provide direct evidence of symmetric vs asymmetric division of HSCs.

Response: We agree with the Reviewer's comments. To examine the stem cell division status under *in vitro* conditions, we stained WT and *Hmga2* KI HSCs by CytoTell to label their plasma membranes and analyzed HSC frequencies after the 2nd and 4th divisions *in vitro* in Figure 1H in the revised manuscript, as shown below. We found that phenotypic HSCs were highly maintained in *Hmga2* KI cells after cell division; however, we did not complete the experiments for cell death and cell homing in transplantation. Nevertheless, to avoid overstating that symmetric division was enhanced by *Hmga2*, we modified these sentences and stopped claiming that *Hmga2* KI HSCs had undergone symmetric division. We instead stated that the overexpression of *Hmga2* drove the expansion of HSCs after the 5-FU injection on page 7 and other pages in the revised manuscript.

3. It is possible that Hmga2 expression is distinct across HSPC populations. Even if its expression was comparable, the chromatin binding site of Hmga2 could be highly cell-type dependent. As the possibility of varied HSPC populations between control and Hmga2 KI at different timepoints of 5FU challenge, it is not certain if the changes observed in Figure 4G-H truly reflect the changes in Hmga2 binding, or it is merely the consequence of different HSPC populations were sampled. It is recommended to quantify Hmga2 expression across various HSPC populations and its binding in the purified HSC population. The same concern also stands for Figure 5F-H.

Response: This is an important question on the distinct binding sites of Hmga2 on chromatin between HSCs and MPP. In response, we performed ChIP-seq analyses of purified HSCs; however, we were unsuccessful, in part, due to the inability to conduct chromatin immunoprecipitation using the antibody on the required number of HSCs. Therefore, we performed ChIP-seq using Lineage-EPCR+ HSPC, which is mostly equivalent to Lineage-c-Kit+Sca1+ (LSK) HSPC. In the near future, we intend to investigate the chromatin-binding sites of Hmga2 in HSCs using an advanced method.

4. Does Casein kinase 2 express in HSCs? Could it be possible that CK2 levels are changed in response to 5FU/stress?

Response: CK2 family genes were all expressed in HSCs showing high TPM values. The expression levels of these genes were not markedly affected in HSCs after the 5-FU injection (please refer to the panels below). In addition, we assessed the phosphorylation of CK2 substrates in the cells of mice after the 5-FU injection using a phospho-CK2 substrate antibody and found that the phosphorylation levels of CK2 substrates were similar before and after the injection. Some substrates were phosphorylated more after the 5-FU injection and TNF- α treatment in the *in vitro* culture, as shown here. We added this information to Appendix Figure S2 in the revised manuscript (please refer to the panels below).

5. It is unclear whether *Hmga2* overexpression was evident in MDS, although GSEA analysis was performed (Figure 6A). *Hmga2* levels should be examined in the age-matched MDS and healthy control HSPCs. This could be easily done using publicly available microarray data.

Response: As suggested by the Reviewer, we examined the expression levels of *HMGA2* in patients with MDS. Although these levels varied among patients, MDS HSPCs generally increased the expression of *HMGA2* in different cohort datasets more than healthy control HSPCs (please refer to the panels below). We added this information to Appendix Figure S3 in the revised manuscript.

6. It will also be helpful to examine *Hmga2* expression over aging in WT mice.

Response: We thank the Reviewer for his/her insightful comment. We compared the expression levels of *Hmga2* in young and old HSCs in published datasets, and observed decreases in old HSCs and HSPCs, as shown here (please refer to the panels below). We added this information to Appendix Figure S3 in the revised manuscript.

Minor concerns:

1. Inconsistent information provided in different sections made it difficult to evaluate some experiments. For instance, tamoxifen treatment to induce *Hmga2* expression in the KI model was described as one month in Result Section 1 (Page 9, line 7-8). However, the same experiment was depicted as 5 consecutive days in the Methods Mice (Page 5, line 6).

Response: We apologize for any confusion. In Figures 1, 2, 4, 5 and 6, we injected mice with tamoxifen on 5 consecutive days and transplanted their mononuclear cells expressing *Hmga2* or lacking the *Hmga2* gene by sorting GFP-positive cells into recipient mice one month after the tamoxifen injection. We added illustrations to clearly show our experimental setting in these figures. Regarding serial transplantation in Supplementary Figure EV1, we injected tamoxifen after bone

marrow transplantation. We corrected the figure legends and added a schematic illustration to the revised manuscript.

2. While no information was provided, it was assumed that no hematopoietic contribution could be observed 12 days post-FU for figures in Figure 1? It will be clear if chimerism can be quantified.

Response: We analyzed CD45.2-positive cells in transplanted Ly5.1+ recipients and showed the data obtained in Figures 1 and 2. We noted that HSC populations mainly comprised 100% CD45.2 cells 12 days after the 5-FU injection, as shown here (please refer to the panels below).

3. Could it be possible to provide targeting (KO) efficiency of Hmga2 KO mouse?

Response: We examined the deletion of Hmga2 by genomic PCR. After the tamoxifen injection in mice, Exons 2 and Exon 3 was not amplified by PCR, as shown in Appendix Figure S1 in the revised manuscript, supporting the deletion of the Hmga2 gene (please refer to the panels below).

4. Are control mice used for Figure 1 and Figure 2 the same? Parameters, including platelet numbers and HSC frequencies, are dramatically different. Could it be possible to provide potential reasons?

Response: As indicated by the Reviewer, these platelet counts and HSCs were somehow different in controls between figures. We repeated the experiments and confirmed the data obtained in these experiments. There were small differences in these counts and frequencies between figures, which may have been caused by as-yet-unidentified factors in the genetic backgrounds and conditions of the mice used.

5. Given that *Hmga2* may suppress inflammatory genes, could this be examined by measuring plasma levels of the same cytokines (Figure 4C) in 5FU-treated *Hmga2*-KI mice?

Response: We analyzed the expression levels of cytokines in plasma in WT and *Hmga2* KI mice after the 5-FU treatment (please refer to the panels below). Cytokine levels slightly decreased in *Hmga2* KI mice. We focused on the role of *Hmga2* in HSCs in the present study, and intend to confirm a reduction in cytokines in bone marrow plasma cells, which may contribute to the quick recovery of hematopoiesis in *Hmga2* KI mice, in the future.

Referee #3:

The authors report an important role for *Hmga2* downstream of CK2 and TNFα in stress hematopoiesis. The story is of interest and the authors need to clarify and better connect a few aspects of the story together as it is outlined below:

Thank you for your time and comments. We performed the additional experiments requested and added the data obtained to the revised manuscript.

A) What are the CK2 levels before and after 5-FU treatment in HSCs? What are the CK2 substrates before and after treatment of HSCs with 5-FU? Is *Hmga2* one of the top substrates? Are other HMG members substrates CK2 substrates as well? Phosphoproteomics analysis should be performed here.

Response: This question was also raised by Reviewer #2, comment 4. The expression levels of CK2 family genes in HSCs were not markedly affected by the 5-FU injection (Please refer to the panel below). We added these data to Appendix Figure S2.

Second: The acquisition of high-quality phosphoproteomic data requires 10 million cells; however, we were unable to collect sufficient numbers of HSPCs. Therefore, pooled lineage-negative BM mononuclear cells were isolated from mice with or without the 5-FU injection and subjected to phosphoproteomic analyses (n=1 each group). The results obtained revealed 2,654

phosphorylated proteins, including the phosphorylation of Hmga2 at Ser-104, one of the 5 potential phosphorylation sites, in samples with or without the 5-FU injection. We added these data to Supplemental Dataset EV6.

Third: Among HMG family proteins, Hmga2, Hmgn3, and Hmgn5 have the canonical CK2 substrate motif, which has glutamic acid or asparagine acid next to the serine residue (Bradley et al. bioRxiv, 2023). Phosphoproteomics revealed the phosphorylation of Hmga1, Hmga2, Hmgn3, Hmgn5, and Hmgb1.

Hmga1	TEKRGRGRPRKQPPVSPG S TALVGSQKEPSEV
Hmgn3	SANGDTKV E EAQR T ES I EKEGE_____
Hmg20a	GRKRKKPLRDSNAPK S PLTGYVRFMNERREQ
Hmgn5	KIEEEGLNEKPGTAK S EDA E VSKD E EEKGD N
Hmga1	GTALVGSQKEPSEVPTPKRPRGRPK S KNKG
Hmga2	QKKPAQETEETSSQ E SA E ED_____
Hmgb1	VQTCREEHKKKHPDA S VNFSEFSK K C S ER W K

B) Can the phenotype observed upon 5-FU (in the case of Hmga2 WT or KI) be rescued by CK2 silencing or the use of CK2 inhibitors?

Response: To answer this question, we injected WT and *Hmga2* KI mice with the CK2 inhibitor, CX4945 six times in three days after the 5-FU injection, and analyzed HSC numbers 7 days after the 5-FU injection. The treatment of 5-FU-injected mice with CX4945 decreased the number of HSCs more in *Hmga2* KI mice than in WT mice, as shown in Figure 5D and 5E in the revised manuscript. These results suggest that the phosphorylation of Hmga2 by CK2 facilitates the expansion of HSCs in mice under stress conditions.

C) How is Hmga2 protein expression regulated (increased) upon 5-FU treatment? Is it through decreased protein turnover via deubiquitination and block of proteasomal degradation?

Response: To elucidate the role of proteasomal degradation in the regulation of Hmga2 protein expression, we analyzed Hmga2 protein expression levels after the 5-FU injection followed by a treatment with either MG132 or cycloheximide. To achieve this, we initially injected 5-FU into *Hmga2* KI mice and collected BM mononuclear cells. These BM cells were cultured with 50 µg/ml cycloheximide or 5 µM MG132 for the indicated times.

The cycloheximide treatment maintained the stability of the Hmga2 protein in cells after the 5-FU injection. However, the MG132 treatment was not able to inhibit the degradation of the Hmga2 protein in control or 5-FU-injected cells, suggesting that another mechanism, such as autophagy, was responsible for Hmga2 protein stability after the 5-FU injection. We intend to investigate the

molecular mechanisms by which the Hmga2 protein was stabilized after the 5-FU treatment in the future.

D) "We next performed RNA sequencing of WT and Hmga2 cKO Lineage-CD150+CD48-EPCR+ HSCs on days 0 and 6 after the 5-FU injection. We identified 1151 up-regulated genes and 717 down-regulated genes in Hmga2 cKO HSCs, relative to those in WT HSCs 6 days after the injection." How does the total amount of RNA isolated from the same number of cells compare between the different sets? I was wondering whether the authors need to use spike-in RNA to normalize their analysis in case the amount of RNA differs a lot.

Response: Difficulties are associated with directly comparing RNA levels in HSCs under different settings because the number of HSCs markedly decreased after the 5-FU injection. Therefore, we intend to semi-quantify RNA levels in HSCs by analyzing the expression of *beta-2-microglobulin* (*B2M*). The expression level of *B2M* is not affected by inflammation, but is dependent on the amount of RNA in specimens (Matsuzaki Y, et al. Regen Ther 2015), as shown below. We compared *B2M* levels in WT or *Hmga2* KO HSCs on days 0 and 6 after the 5-FU treatment. No significant changes were observed between each group (please refer to the panels below).

E) Can ectopic expression of RFX5 rescue the Hmga2-mediated phenotype?

Response: Since the RFX5 binding motif was enriched in the Hmga2-5A mutant after the TNF- α treatment, as shown in Figure 6F in the revised manuscript, we transduced murine Rfx5- and Rfx3- expressing vectors into WT or *Hmga2* KI HSCs and treated them with TNF- α for three days in a liquid culture. We found that the overexpression of Rfx5 decreased the number of HSCs after the TNF- α treatment significantly more than the control vector (please refer to the panels below). We added this interesting result to Figure 6K and 6L on page 15 in the revised manuscript.

F) The connection of the findings with MDS is rather loose. The authors need to discuss more about specific genes, including RFX5.

The paper by Moison et al., *Blood Adv.* 2022 Aug 23 (AML setting) might be an interesting study and the authors could compare their gene expression signatures to the signatures in this paper.

Response: We thank the Reviewer for this comment. We analyzed the *Hmga2*-dependent transcriptome in human AML by using the datasets of GSE15061 (Moison, et al. *Blood Advances* 2022), and found that *Hmga2*-high AML cells significantly increased the expression of genes that were up-regulated in 5-FU-treated *Hmga2* KI HSCs, but did not change the expression of those that were down-regulated, compared to *Hmga2* null AML cells. GSEA results on AML are shown in Figure 7A. This observation in AML suggests that *Hmga2* repressed the expression of genes in AML HSPCs in a distinct manner from the stress-induced activation of *Hmga2* in normal HSCs.

In addition, we examined the expression levels of the *RFX5* gene in MDS cells, and found that *RFX5* mRNA expression was slightly higher in MDS cells than in healthy controls in some published datasets (please refer to the panels below); therefore, the underlying mechanisms for MDS progression will be investigated in the future. Overall, in response to the comments by Reviewers #1 and 3, we changed the relevant sentences to soften our statements on the role of *Hmga2* in MDS on page 15-16 in the revised manuscript.

Dear Dr Goro Shashida,

Thank you for submitting your revised manuscript (EMBOJ-2023-115005R) to The EMBO Journal. Your amended study was sent back to the three referees for their scientific re-evaluation, and we have received detailed comments from all of them, which I enclose below. As you will see, the experts state that the work has been substantially improved by the revisions and they are now broadly in favour of publication.

Thus, we are pleased to inform you that your manuscript has been accepted in principle for publication in The EMBO Journal.

We now need you to take care of a number of issues related to formatting and data presentation as detailed below, which should be addressed at re-submission.

Please contact me at any time if you have additional questions related to below points.

As you might have seen on our web page, every paper at the EMBO Journal now includes a 'Synopsis', displayed on the html and freely accessible to all readers. The synopsis includes a 'model' figure as well as 2-5 one-short-sentence bullet points that summarize the article. I would appreciate if you could provide this figure and the bullet points.

Thank you for giving us the chance to consider your manuscript for The EMBO Journal. I look forward to your final revision.

Again, please contact me at any time if you need any help or have further questions.

Best regards,

Daniel Klimmeck

>> Please add up to five keywords for your study.

>> Add a 'Disclosure and Competing Interests Statement' to the main article text.

>> Callouts: complement with callouts for Figures EV1-4.

>> Funding: currently missing in our online system 'Japanese Society of Hematology to Goro Sashida' and '22KJ2522'. Are the grant numbers available for the other funders (Takeda Science Foundation (TSF) etc?).

>> Figures should be uploaded in TIFF, EPS or PDF format.

>> Appendix: appendix figures should be compiled in one PDF labelled "Appendix". The files should include a table of contents with page numbers, and the figures together with their legends. The legends should be removed from the manuscript.

>> Dataset EV Legends: please double-check the file numbering for Dataset EV8/9. The legends should be removed from the manuscript text and added to the corresponding datasets.

>> Source Data: raw data for western blots for Fig 5B,C, Fig 6A should be removed from the appendix and uploaded as separate source data files. Files should be uploaded as one file per figure.

>> Remove the 'data not shown' statement on p.19, or add respective data.

>> Please cite referencing your previous J Exp Med study (PMID: 37071125) in the Material and Methods section.

>> Move the Proteomics methods annotation from EV6 to the main material & methods part.

>> Data Availability Section: please move sequencing dataset IDs to the 'Data Availability Section'. Move data annotation from EV9 to the main text Material and Methods.

>> Consider additional changes and comments from our production team as indicated below:

- DAS: Please note that the accession ID for the DDBJ database is not provided in the data availability statement.

- Figure legends:

1. Please note that a separate 'Data Information' section is required in the legends of figures 1b-h, j; 2b, d-h; 4a-f, h-i; 6i- j, l; EV 1b, d.

2. Please note that the figure EV 1b, d does not contain any statistical parameter, kindly rectify the statistical test and p-value related information in the figure legend appropriately.

3. Please indicate the statistical test used for data analysis in the legends of figures 3c, e; 4l, n; 6f; EV 4a; EV 5a-c.

4. Please note that information related to n is missing in the legend of figures EV 2b.

5. Although 'n' is provided, please describe the nature of entity for 'n' in the legends of figures 1b, d, f, h, j; 4a-f; 6e, i-j, l; EV 1b, d; EV 3a-f.

6. Please note that the error bars are not defined in the legend of figure EV 2b.

7. Please note that for heatmap present in figures 4j-k; EV 4b; a numbered scale bar is not provided. This needs to be rectified.

Referee #1:

Thank you to the authors for the effort they have made to respond to my questions and concerns. The addition of the new experiments, additional data points and text adaptations have, in my opinion, greatly improved the manuscript. Especially the addition of in vivo TNFa treatments add an additional form of stress to the manuscript, making it less focused just on 5-FU. While going through the manuscript again, it looks like Hmga2 might be a general player in controlling increased proliferation in stress conditions, such as chemotherapy treatment, transplantation or cytokine treatment. Therefore I do still have some concern that all mouse experiments are performed on a transplantation background. It would have been nice to see data from experiments in which the time between transplant, tamoxifen treatment and 5-FU would have been a bit further apart for the system to recover better from initial transplant stress.

Minor point 2, my last question on wt HSCs has not been addressed in their response. Would still be interesting to know.

Referee #2:

After a careful review of the modified manuscript, it is believed that the authors have attempted to improve the manuscript with their best efforts albeit some challenges in addressing all points due to technical limitations. Therefore, it is recommended for publication.

Referee #3:

The authors have sufficiently addressed all my comments. The manuscript reports significant findings in the field of HSC biology and response to stress.

Referee #1:

Thank you to the authors for the effort they have made to respond to my questions and concerns. The addition of the new experiments, additional data points and text adaptations have, in my opinion, greatly improved the manuscript. Especially the addition of in vivo TNF α treatments add an additional form of stress to the manuscript, making it less focused just on 5-FU. While going through the manuscript again, it looks like Hmga2 might be a general player in controlling increased proliferation in stress conditions, such as chemotherapy treatment, transplantation or cytokine treatment. Therefore I do still have some concern that all mouse experiments are performed on a transplantation background. It would have been nice to see data from experiments in which the time between transplant, tamoxifen treatment and 5-FU would have been a bit further apart for the system to recover better from initial transplant stress. Minor point 2, my last question on wt HSCs has not been addressed in their response. Would still be interesting to know.

Response: We thank the reviewer for his/her time and comments, which have improved our manuscript. As the reviewer has a concern about our experimental setting, we would examine how Hmga2 contributes to hematopoietic recovery in response to stresses in primary mice using a blood-specific Cre system in future.

Referee #2:

After a careful review of the modified manuscript, it is believed that the authors have attempted to improve the manuscript with their best efforts albeit some challenges in addressing all points due to technical limitations. Therefore, it is recommended for publication.

Response: We thank the reviewer for his/her time and comments, which have improved our manuscript.

Referee #3:

The authors have sufficiently addressed all my comments. The manuscript reports significant findings in the field of HSC biology and response to stress.

Response: We thank the reviewer for his/her time and comments, which have improved our manuscript.

Dear Dr Goro Sashida,

Thank you for submitting the revised version of your manuscript. I have now evaluated your amended manuscript and concluded that the remaining minor concerns have been sufficiently addressed.

I am pleased to inform you that your manuscript has been accepted for publication in the EMBO Journal.

On a different note, I would like to alert you that EMBO Press offers a format for a video-synopsis of work published with us, which essentially is a short, author-generated film explaining the core findings in hand drawings, and, as we believe, can be very useful to increase visibility of the work. Please see the following link for representative examples and their integration into the article web page:

<https://www.embopress.org/doi/full/10.15252/emboj.2019103932>

Kind regards,

Daniel Klimmeck

Daniel Klimmeck, PhD
Senior Editor
The EMBO Journal
EMBO
Postfach 1022-40
Meyerhofstrasse 1
D-69117 Heidelberg
contact@embojournal.org
Submit at: <http://emboj.msubmit.net>
